# GRAPH NEURAL NETWORKS WITH LEARNABLE STRUCTURAL AND POSITIONAL REPRESENTATIONS

**Vijay Prakash Dwivedi**[1]
vijaypra001@e.ntu.edu.sg

**Anh Tuan Luu**[1]
anhtuan.luu@ntu.edu.sg

**Thomas Laurent**[2]
tlaurent@lmu.edu

**Yoshua Bengio**[3,4]
yoshua.bengio@mila.quebec

**Xavier Bresson**[5]
xavier@nus.edu.sg

[1] Nanyang Technological University, Singapore   [2] Loyola Marymount University
[3] Mila, University of Montréal   [4] CIFAR   [5] National University of Singapore

## ABSTRACT

Graph neural networks (GNNs) have become the standard learning architectures for graphs. GNNs have been applied to numerous domains ranging from quantum chemistry, recommender systems to knowledge graphs and natural language processing. A major issue with arbitrary graphs is the absence of canonical positional information of nodes, which decreases the representation power of GNNs to distinguish e.g. isomorphic nodes and other graph symmetries. An approach to tackle this issue is to introduce Positional Encoding (PE) of nodes, and inject it into the input layer, like in Transformers. Possible graph PE are Laplacian eigenvectors. In this work, we propose to decouple structural and positional representations to make easy for the network to learn these two essential properties. We introduce a novel generic architecture which we call LSPE (Learnable Structural and Positional Encodings). We investigate several sparse and fully-connected (Transformer-like) GNNs, and observe a performance increase for molecular datasets, from $1.79\%$ up to $64.14\%$ when considering learnable PE for both GNN classes. [1]

## 1 INTRODUCTION

GNNs have recently emerged as a powerful class of deep learning architectures to analyze datasets where information is present in the form of heteregeneous graphs that encode complex data connectivity. Experimentally, these architectures have shown great promises to be impactful in diverse domains such as drug design (Stokes et al., 2020; Gaudelet et al., 2020), social networks (Monti et al., 2019; Pal et al., 2020), traffic networks (Derrow-Pinion et al., 2021), physics (Cranmer et al., 2019; Bapst et al., 2020), combinatorial optimization (Bengio et al., 2021; Cappart et al., 2021) and medical diagnosis (Li et al., 2020c).

Most GNNs (such as Defferrard et al. (2016); Sukhbaatar et al. (2016); Kipf & Welling (2017); Hamilton et al. (2017); Monti et al. (2017); Bresson & Laurent (2017); Veličković et al. (2018); Xu et al. (2019)) are designed with a message-passing mechanism (Gilmer et al., 2017) that builds node representation by aggregating local neighborhood information. It means that this class of GNNs is fundamentally structural, i.e. the node representation only depends on the local structure of the graph. As such, two atoms in a molecule with the same neighborhood are expected to have similar representation. However, it can be limiting to have the same representation for these two atoms as their positions in the molecule are distinct, and their role may be specifically separate (Murphy et al., 2019). As a consequence, the popular message-passing GNNs (MP-GNNs) fail to differentiate two nodes with the same 1-hop local structure. This restriction is now properly understood in the context of the equivalence of MP-GNNs with Weisfeiler-Leman (WL) test (Weisfeiler & Leman, 1968) for graph isomorphism (Xu et al., 2019; Morris et al., 2019).

---

[1]Code: https://github.com/vijaydwivedi75/gnn-lspe

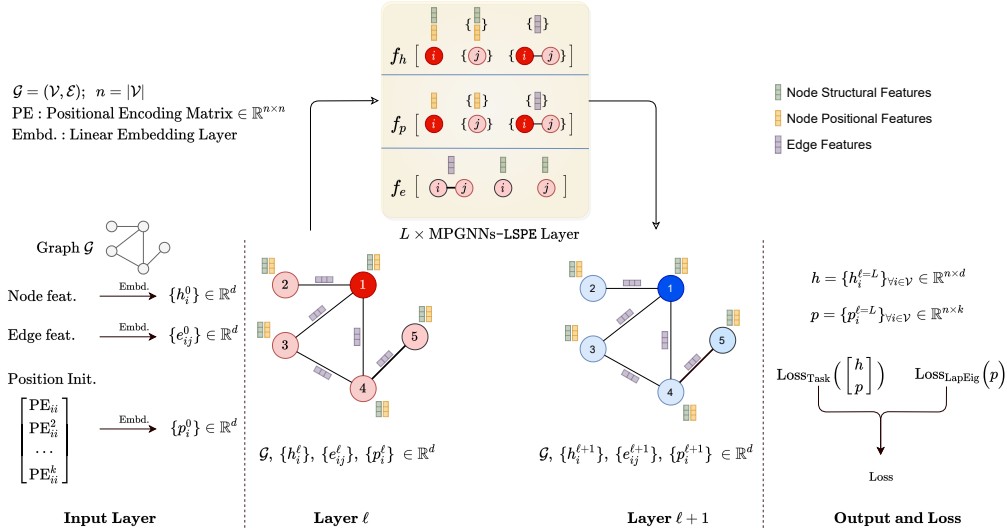

Figure 1: Block diagram illustration of the proposed MPGNNs-LSPE architecture along with the inputs, general framework of a layer, and the output and loss components.

The said limitation can be alleviated, to certain extents, by (i) stacking multiple layers, (ii) applying higher-order GNNs, or (iii) considering positional encoding (PE) of nodes (and edges). Let us assume two structurally identical nodes in a graph with the same 1-hop neighborhood, but different with respect to 2-hop or higher-order neighborhoods. Then, stacking several layers (Bresson & Laurent, 2017; Li et al., 2019) can propagate the information from a node to multiple hops, and thus differentiate the representation of two far-away nodes. However, this solution can be deficient for long-distance nodes because of the over-squashing phenomenon (Alon & Yahav, 2020). Another approach is to compute higher-order node-tuple aggregations such as in WL-based GNNs (Maron et al., 2019; Chen et al., 2019); though these models are computationally more expensive to scale than MP-GNNs, even for medium-sized graphs (Dwivedi et al., 2020). An alternative technique is to consider a global positioning of the nodes in the graph that can encode a graph-based distance between the nodes (You et al., 2019; Dwivedi et al., 2020; Li et al., 2020b; Dwivedi & Bresson, 2021), or can inform about specific sub-structures (Bouritsas et al., 2020; Bodnar et al., 2021).

**Contribution.** In this work, we turn to the idea of *learning positional representation* that can be combined with structural GNNs to generate more expressive node embedding. Our main intent is to alleviate the lack of canonical positioning of nodes in arbitrary graphs to improve the representation power of MP-GNNs, while keeping their linear complexity for large-scale applications. For this objective, we propose a novel framework, illustrated with Figure 1, that enables GNNs to learn both structural and positional representations at the same time (thus named MPGNNs-LSPE). Alongside, we present a random-walk diffusion based positional encoding scheme to initialize the positional representations of the nodes. We show that the proposed architecture with learnable PE can be used with any graph network that fits to the MP-GNNs framework, and improves its performance (1.79% to 64.14%). In our demonstrations, we formulate LSPE instances of both sparse GNNs, such as GatedGCNs (Bresson & Laurent, 2017) and PNA (Corso et al., 2020) and fully-connected Transformers-based GNNs (Kreuzer et al., 2021; Mialon et al., 2021). Our numerical experiments on three standard molecular benchmarks show that different instantiations of MP-GNNs with LSPE surpass the previous state-of-the-art (SOTA) on one dataset by a considerable margin (26.23%), while achieving SOTA-comparable score on the other two datasets. The architecture also shows consistent improvements on three non-molecular benchmarks. In addition, our evaluations find the sparse MP-GNNs to be outperforming fully-connected GNNs, hence suggesting greater potential towards the development of highly efficient, yet powerful architectures for graphs.

## 2 RELATED WORK

In this section, we review briefly the three research directions theoretical expressivity of GNNs, graph positional encoding, and Transformer-based GNNs.

**Theoretical expressivity and Weisfeiler-Leman GNNs.** As the theoretical expressiveness of MP-GNNs is bounded by the 1-WL test (Xu et al., 2019; Morris et al., 2019), they may perform poorly on graphs that exhibit several symmetries (Murphy et al., 2019), and additionally some message-passing functions may not be discriminative enough (Corso et al., 2020). To this end, $k$-order Equivariant-GNNs were introduced in Maron et al. (2018) requiring $O(n^k)$ memory and speed complexities. Although the complexity was improved to $O(n^2)$ memory and $O(n^3)$ respectively (Maron et al., 2019; Chen et al., 2019; Azizian & Lelarge, 2020), it is still inefficient compared with the linear complexity of MP-GNNs.

**Graph Positional Encoding.** The idea of positional encoding, i.e. the notion of global position of pixels in images, words in texts and nodes in graphs, plays a central role in the effectiveness of the most prominent neural networks with ConvNets (LeCun et al., 1998), RNNs (Hochreiter & Schmidhuber, 1997), and Transformers (Vaswani et al., 2017). For GNNs, the position of nodes is more challenging due to the fact that there does not exist a canonical positioning of nodes in arbitrary graphs. Despite these issues, graph positional encoding are as much critical for GNNs as they are for ConvNets, RNNs and Transformers, as demonstrated for prediction tasks on graphs (Srinivasan & Ribeiro, 2019; Cui et al., 2021). Nodes in a graph can be assigned index positional encoding (PE). However, such a model must be trained with the $n!$ possible index permutations or else sampling needs to be done (Murphy et al., 2019). Another PE candidate for graphs can be Laplacian Eigenvectors (Dwivedi et al., 2020; Dwivedi & Bresson, 2021) as they form a meaningful local coordinate system, while preserving the global graph structure. However, there exists sign ambiguity in such PE as eigenvectors are defined up to $\pm 1$, leading to $2^k$ number of possible sign values when selecting $k$ eigenvectors which a network needs to learn. Similarly, the eigenvectors may be unstable due to eigenvalue multiplicities. You et al. (2019) proposed learnable position-aware embeddings based on random anchor sets of nodes, where the random selection of anchors has its limitations, which makes their approach less generalizable on inductive tasks. There also exists methods that encode prior information about a class of graphs of interest such as rings for molecules (Bouritsas et al., 2020; Bodnar et al., 2021) which make MP-GNNs more expressive. But the prior information regarding graph sub-structures needs to be pre-computed, and sub-graph matching and counting require $O(n^k)$ for $k$-tuple sub-structure.

**Transformer-based GNNs.** Although sparse MP-GNNs are very efficient, they are susceptible to the information bottleneck limitation (Alon & Yahav, 2020) in addition to vanishing gradient (similar to RNNs) on tasks when long-range interactions between far away nodes are critical. To overcome these limitations, there have been recent works that generalize Transformers to graphs (Dwivedi & Bresson, 2021; Kreuzer et al., 2021; Ying et al., 2021; Mialon et al., 2021) which alleviates the long-range issue as 'everything is connected to everything'. However, these methods either use non-learnable PEs to encode graph structure information (Dwivedi & Bresson, 2021; Ying et al., 2021; Mialon et al., 2021), or inject learned PEs to the Transformer network that relies on Laplacian eigenvectors (Kreuzer et al., 2021), thus inheriting the sign ambiguity limitation.

A detailed review of the above research directions is available in the supplementary Section B. We attempt to address some of the major limitations of GNNs by proposing a novel architecture with consistent performance gains.

## 3 PROPOSED ARCHITECTURE

In this work, we decouple structural and positional representations to make it easy for the network to learn these two critical characteristics. This is in contrast with most existing architectures s.a. Dwivedi & Bresson (2021); Beani et al. (2021); Kreuzer et al. (2021) that inject the positional information into the input layer of the GNNs, and You et al. (2019) that rely on distance-measured anchor sets of nodes limiting general, inductive usage. Given the recent theoretical results on the importance of informative graph PE for expressive GNNs (Murphy et al., 2019; Srinivasan & Ribeiro, 2019; Loukas, 2020), we are interested in a generic framework that can enable GNNs to separate positional and structural representations to increase their expressivity. Section 3.1 will introduce our approach to augment GNNs with learnable graph PE. Our framework can be used with different GNN architectures. We illustrate this flexibility in Sections C.1 and C.2 where the decoupling of structural and positional information is applied to both sparse MP-GNNs and fully-connected GNNs.

## 3.1 Generic Formulation: MP-GNNs–LSPE

**Notation.** Let $\mathcal{G} = (\mathcal{V}, \mathcal{E})$ be a graph with $\mathcal{V}$ being the set of nodes and $\mathcal{E}$ the set of edges. The graph has $n = |\mathcal{V}|$ nodes and $E = |\mathcal{E}|$ edges. The connectivity of the graph is represented by the adjacency matrix $A \in \mathbb{R}^{n \times n}$ where $A_{ij} = 1$ if there exists an edge between the nodes $i$ and $j$; otherwise $A_{ij} = 0$. The degree matrix is denoted $D \in \mathbb{R}^{n \times n}$. The node features and positional features for node $i$ is denoted by $h_i$ and $p_i$ respectively, while the features for an edge between nodes $i$ and $j$ is indicated by $e_{ij}$. A GNN model is composed of three main components; an embedding layer for the input features, a stack of convolutional layers, and a final task-based layer, as in Figure 1. The layers are indexed by $\ell$ and $\ell = 0$ denotes the input layer.

**Standard MP-GNNs.** Considering a graph which has available node and edge features, and these are transformed at each layer, the update equations for a conventional MP-GNN layer are defined as:

$$\text{MP-GNNs}: \quad h_i^{\ell+1} \quad = \quad f_h\Big(h_i^\ell, \{h_j^\ell\}_{j \in \mathcal{N}_i}, e_{ij}^\ell\Big),\ h_i^{\ell+1}, h_i^\ell \in \mathbb{R}^d, \tag{1}$$

$$e_{ij}^{\ell+1} \quad = \quad f_e\Big(h_i^\ell, h_j^\ell, e_{ij}^\ell\Big),\ e_{ij}^{\ell+1}, e_{ij}^\ell \in \mathbb{R}^d, \tag{2}$$

where $f_h$ and $f_e$ are functions with learnable parameters, and $\mathcal{N}_i$ is the neighborhood of the node $i$. The design of functions $f_h$ and $f_e$ depends on the GNN architecture used, see Zhou et al. (2020) for a review. As Transformer neural networks (Vaswani et al., 2017) are a special case of MP-GNNs (Joshi, 2020), Eq. (1) can be simplified to encompass the original Transformers by dropping the edge features and making the graph fully connected.

**Input features and initialization.** The node and edge features at layer $\ell = 0$ are produced by a linear embedding of available input node and edge features denoted respectively by $h_i^{\text{in}} \in \mathbb{R}^{d_v}, e_{ij}^{\text{in}} \in \mathbb{R}^{d_e}$: $h_i^{\ell=0} = \text{LL}_h(h_i^{\text{in}}) = A^0 h_i^{\text{in}} + a^0 \in \mathbb{R}^d, e_{ij}^{\ell=0} = \text{LL}_e(e_{ij}^{\text{in}}) = B^0 e_{ij}^{\text{in}} + b^0 \in \mathbb{R}^d$, where $A^0 \in \mathbb{R}^{d \times d_v}$, $B^0 \in \mathbb{R}^{d \times d_e}$ and $a^0, b^0 \in \mathbb{R}^d$ are the learnable parameters of the linear layers.

**Positional Encoding.** Existing MP-GNNs that integrate positional information usually propose to concatenate the PE with the input node features, similarly to Transformers (Vaswani et al., 2017):

$$\text{MP-GNNs-PE}: \quad h_i^{\ell+1} = f_h\Big(h_i^\ell, \{h_j^\ell\}_{j \in \mathcal{N}_i}, e_{ij}^\ell\Big),\ h_i^{\ell+1}, h_i^\ell \in \mathbb{R}^d, \tag{3}$$

$$e_{ij}^{\ell+1} = f_e\Big(h_i^\ell, h_j^\ell, e_{ij}^\ell\Big),\ e_{ij}^{\ell+1}, e_{ij}^\ell \in \mathbb{R}^d, \tag{4}$$

$$\text{with initial } h_i^{\ell=0} = \text{LL}_h\left(\begin{bmatrix} h_i^{\text{in}} \\ p_i^{\text{in}} \end{bmatrix}\right) = D^0 \begin{bmatrix} h_i^{\text{in}} \\ p_i^{\text{in}} \end{bmatrix} + d^0 \in \mathbb{R}^d, \tag{5}$$

$$\text{and } e_{ij}^{\ell=0} = \text{LL}_e(e_{ij}^{\text{in}}) = B^0 e_{ij}^{\text{in}} + b^0 \in \mathbb{R}^d, \tag{6}$$

where $p_i^{\text{in}} \in \mathbb{R}^k$ is the input PE of node $i$, $D^0 \in \mathbb{R}^{d \times (d_v + k)}, d^0 \in \mathbb{R}^d$ are parameters for the linear transformation. Such architecture merges the positional and structural representations together. It has the advantage to keep the same linear complexity for learning, but it does not allow the positional representation to be changed and better adjusted to the task at hand.

**Decoupling position and structure in MP-GNNs.** We decouple the positional information from the structural information such that both representations are learned separately resulting in an architecture with **L**earnable **S**tructural and **P**ositional **E**ncodings, which we call **MP-GNNs–LSPE**. The layer update equations are defined as:

$$\text{MP-GNNs-LSPE}: \quad h_i^{\ell+1} = f_h\left(\begin{bmatrix} h_i^\ell \\ p_i^\ell \end{bmatrix}, \left\{\begin{bmatrix} h_j^\ell \\ p_j^\ell \end{bmatrix}\right\}_{j \in \mathcal{N}_i}, e_{ij}^\ell\right),\ h_i^{\ell+1}, h_i^\ell \in \mathbb{R}^d, \tag{7}$$

$$e_{ij}^{\ell+1} = f_e\Big(h_i^\ell, h_j^\ell, e_{ij}^\ell\Big),\ e_{ij}^{\ell+1}, e_{ij}^\ell \in \mathbb{R}^d, \tag{8}$$

$$p_i^{\ell+1} = f_p\Big(p_i^\ell, \{p_j^\ell\}_{j \in \mathcal{N}_i}, e_{ij}^\ell\Big),\ p_i^{\ell+1}, p_i^\ell \in \mathbb{R}^d, \tag{9}$$

The difference of this architecture with the standard MP-GNNs is the addition of the positional representation update Eq. (9), along with the concatenation of these learnable PEs with the node structural features, Eq. (7). As we will see in the next section, the design of the message-passing function $f_p$ follows the same analytical form of $f_h$ but with the use of the tanh activation function to allow positive and negative values for the positional coordinates. It should be noted that the inclusion

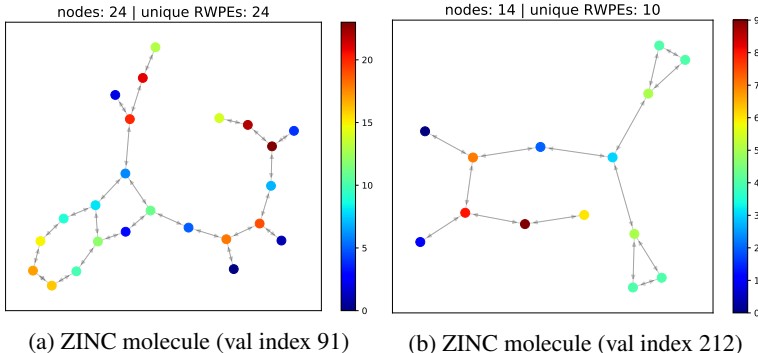

(a) ZINC molecule (val index 91)       (b) ZINC molecule (val index 212)

Figure 2: Sample graph plots from the ZINC validation set with each node color in a graph representing a unique RWPE vector, when $k = 24$. The corresponding graph ids, the number of nodes in the graphs and the number of unique RWPEs are labelled against the figures.

of the edge features, $e_{ij}^{\ell}$ in the $h$ or $p$ update is optional as several MP-GNNs do not include edge features in their $h$ updates. Nevertheless, the architecture we present is made as generic so as to be used for future extensions in a convenient way.

**Definition of initial PE.** The choice of the initial PE is critical. In this work, we consider two PEs: Laplacian PE (LapPE) and Random Walk PE (RWPE). LapPE are defined in Section B.2 as $p_i^{\text{LapPE}} = [\ U_{i1}, U_{i2}, \cdots, U_{ik}\ ] \in \mathbb{R}^k$. LapPE provide a unique node representation and are distance-sensitive w.r.t. the Euclidean norm. However, they are limited by the sign ambiguity, which requires random sign flipping during training for the network to learn this invariance (Dwivedi et al., 2020).

Inspired by Li et al. (2020b), we propose RWPE, a PE based on the random walk (RW) diffusion process (although other graph diffusions can be considered s.a. PageRank (Mialon et al., 2021)). Formally, RWPE are defined with $k$-steps of random walk as:

$$p_i^{\text{RWPE}} \quad = \quad [\ \text{RW}_{ii}, \text{RW}_{ii}^2, \cdots, \text{RW}_{ii}^k\ ] \in \mathbb{R}^k, \tag{10}$$

where $\text{RW} = AD^{-1}$ is the random walk operator. In contrast of Li et al. (2020b) which uses the full matrix $\text{RW}_{ij}$ for all pairwise nodes, we adopt a low-complexity usage of the random walk matrix by considering only the landing probability of a node $i$ to itself, i.e. $\text{RW}_{ii}$. Note that these PE do not suffer from the sign ambiguity of LapPE, so the network is not required to learn additional invariance. RWPE provide a unique node representation under the condition that each node has a unique $k$-hop topological neighborhood for a sufficient large $k$. This assumption can be discussed. If we consider synthetic strongly regular graphs like the CSL graphs (Murphy et al., 2019), then all nodes in a graph have the same RWPE for any $k$ value, since they are isomorphic by construction. However, despite RWPE being the same for all nodes in a graph, these PE are unique for each class of isomorphic graphs, resulting in a perfect classification of the CSL dataset, see Section A.1. For graphs such as Decalin and Bicyclopentyl (Sato, 2020), nodes which are not isomorphic receive different RWPE for $k \geq 5$, also in Section A.1. Finally, for real-world graphs like ZINC molecules, most nodes receive a unique node representation for $k \geq 24$, see Figure 2 for an illustration, where the two molecules have $100\%$ and $71.43\%$ unique RWPEs respectively. Section A.3 presents a detailed study.

Experimentally, we will show that RWPE outperform LapPE, suggesting that learning the sign invariance is more difficult (as there exist $2^k$ possible sign flips for each graph) than not exactly having unique node representation for each node. As mentioned above for CSL, RWPE are related to the problem of graph isomorphism and higher-order node interactions. Precisely, iterating the random walk operator for a suitable number of steps allows coloring non-isomorphic nodes, thus distinguishing several cases of non-isomorphic graphs on which the 1-WL test, and equivalently MP-GNNs, fail s.a. the CSL, Decalin and Bicyclopentyl graphs. We refer to Section A.2 for a formal presentation of the iterative algorithm. Finally, the initial PE of the network is obtained by embedding the LapPE or RWPE into a $d$-dimensional feature vector:

$$p_i^{\ell=0} \quad = \quad \text{LL}_p(p_i^{\text{PE}}) = C^0 p_i^{\text{PE}} + c^0 \in \mathbb{R}^d, \qquad \text{where } C^0 \in \mathbb{R}^{d \times k}, c^0 \in \mathbb{R}^d. \tag{11}$$

**Positional loss.** As we separate the learning of the structual and positional representations, it is possible to consider a specific positional encoding loss along with the task loss. A natural candidate

is the Laplacian eigenvector loss (Belkin & Niyogi, 2003; Lai & Osher, 2014) that enforces the PE to form a coordinate system constrained by the graph topology. As such, the final loss function of **MP-GNNs−LSPE** is composed of two terms:

$$\text{Loss} = \text{Loss}_{\text{Task}}\left(\begin{bmatrix} h^{\ell=L} \\ p^{\ell=L} \end{bmatrix}\right) + \alpha \, \text{Loss}_{\text{LapEig}}(p^{\ell=L}), \tag{12}$$

where $h^{\ell=L} \in \mathbb{R}^{n \times d}, p^{\ell=L} \in \mathbb{R}^{n \times k}$, $k$ is the dimension of learned PE, $\ell = L$ is the final GNN layer, and $\alpha > 0$ an hyper-parameter. Observe also that we enforce the final positional vectors $p^{\ell=L}$ to have centered and unit norm with $\text{mean}(p^{\ell=L}_{.,k}) = 0$, $\|p^{\ell=L}_{.,k}\| = 1$, $\forall k$ to better approximate the Laplacian eigenvector loss defined by $\text{Loss}_{\text{LapEig}}(p) = \frac{1}{k} \, \text{trace}\big(p^T \Delta p\big) + \frac{\lambda}{k} \, \big\|p^T p - \text{I}_k\big\|_F^2$ with $\lambda > 0$ and $\| \cdot \|_F^2$ being the Frobenius norm.

## 3.2 INSTANCES OF LSPE WITH MP-GNNS AND TRANSFORMER GNNS

We instantiate two classes of GNN architectures, both *sparse* MP-GNNs and *fully-connected* Transformer GNNs using our proposed LSPE framework. For sparse MP-GNNs, we consider GatedGCN (Bresson & Laurent, 2017) and PNA (Corso et al., 2020), while we extend the recently developed SAN (Kreuzer et al., 2021) and GraphiT (Mialon et al., 2021) with LSPE to develop Transformer-LSPE architectures. We briefly demonstrate here how a GNN can be instantiated using LSPE (Eqs. (7-9)) by developing GatedGCN−LSPE (Eqs. (14-16)), while the complete equations for the four models are defined in Section C of the supplementary material, given the space constraint.

**GatedGCN−LSPE:** Originally, GatedGCNs are sparse MP-GNNs equipped with a soft-attention mechanism that is able to learn adaptive edge gates to improve the message aggregation step of GCN networks (Kipf & Welling, 2017). Our proposed extension of this model with LSPE is defined as:

$$h^{\ell+1}, e^{\ell+1}, p^{\ell+1} = \text{GatedGCN−LSPE}\Big(h^\ell, e^\ell, p^\ell\Big), \; h \in \mathbb{R}^{n \times d}, e \in \mathbb{R}^{E \times d}, p \in \mathbb{R}^{n \times d}, \tag{13}$$

$$\text{with } h_i^{\ell+1} = h_i^\ell + \text{ReLU}\Big(\text{BN}\Big(A_1^\ell \begin{bmatrix} h_i^\ell \\ p_i^\ell \end{bmatrix} + \sum_{j \in \mathcal{N}(i)} \eta_{ij}^\ell \odot A_2^\ell \begin{bmatrix} h_j^\ell \\ p_j^\ell \end{bmatrix}\Big)\Big), \tag{14}$$

$$e_{ij}^{\ell+1} = e_{ij}^\ell + \text{ReLU}\big(\text{BN}\big(\hat{\eta}_{ij}^\ell\big)\big), \tag{15}$$

$$p_i^{\ell+1} = p_i^\ell + \tanh\Big(C_1^\ell p_i^\ell + \sum_{j \in \mathcal{N}(i)} \eta_{ij}^\ell \odot C_2^\ell p_j^\ell\Big), \tag{16}$$

where $\eta_{ij}^\ell = \sigma\big(\hat{\eta}_{ij}^\ell\big)/\big(\sum_{j' \in \mathcal{N}(i)} \sigma\big(\hat{\eta}_{ij'}^\ell\big) + \epsilon\big), \hat{\eta}_{ij}^\ell = B_1^\ell h_i^\ell + B_2^\ell h_j^\ell + B_3^\ell e_{ij}^\ell, h_i^\ell, e_{ij}^\ell, p_i^\ell, \eta_{ij}^\ell, \hat{\eta}_{ij}^\ell \in \mathbb{R}^d, A_1^\ell, A_2^\ell \in \mathbb{R}^{d \times 2d}$ and $B_1^\ell, B_2^\ell, B_3^\ell, C_1^\ell, C_2^\ell \in \mathbb{R}^{d \times d}$. Notice the $p$-update in Eq. (16) follows the same analytical form as the $h$-update in Eq. (14) except for the difference in activation function, and omission of BN, which was not needed in our experiments.

## 4 NUMERICAL EXPERIMENTS

We evaluate the proposed MPGNNs−LSPE architecture on the instances of sparse GNNs and Transformer GNNs defined in Section 3.2 (all models are presented in Section C), using PyTorch (Paszke et al., 2019) and DGL (Wang et al., 2019) on standard molecular benchmarks – ZINC (Irwin et al., 2012), OGBG-MOLTOX21 and OGBG-MOLPCBA (Hu et al., 2020). ZINC and MOLTOX21 are of medium scale with 12K and 7.8K graphs respectively, whereas MOLPCBA is of large scale with 437.9K graphs. These datasets, each having a global graph-level property to be predicted, consist of molecules which are represented as graphs of atoms as nodes and bonds between the atoms as edges. Additionally, we evaluate our architecture on three non-molecular graph datasets to show the usefulness of LSPE on any graph domain in general, see Section D in the supplementary.

### 4.1 DATASETS AND EXPERIMENTAL SETTINGS

**ZINC** is a graph regression dataset where the property to be predicted for a graph is its constrained solubility which is a vital chemical property in molecular design (Jin et al., 2018). We use the 12,000 subset of the dataset with the same splitting defined in Dwivedi et al. (2020). Mean Absolute Error

(MAE) of the property being regressed is the evaluation metric. **OGBG-MOLTOX21** is a multi-task binary graph classification dataset where a qualitative (active/inactive) binary label is predicted against 12 different toxicity measurements for each molecular graph (Tox21, 2014; Wu et al., 2018). We use the scaffold-split version of the dataset included in OGB (Hu et al., 2020) that consists of 7,831 graphs. ROC-AUC averaged across the tasks is the evaluation metric. **OGBG-MOLPCBA** is also a multi-task binary graph classification dataset from OGB where an active/inactive binary label is predicted for 128 bioassays (Wang et al., 2012; Wu et al., 2018). It has 437,929 graphs with scaffold-split and the evaluation metric is Average Precision (AP) averaged over the tasks.

To evaluate different instantiations of our proposed MPGNNs-LSPE, we follow the same benchmarking protocol in Dwivedi et al. (2020) to fairly compare several models on a fixed number of 500k model parameters, for ZINC. We relax the model sizes to larger parameters for evaluation on the two OGB datasets as observed being practised on their leaderboards (Hu et al., 2020). The total size of parameters of each model, including the number of layers used, are indicated in the respective experiment tables, with the remaining implementation details included in supplementary Section E.

## 4.2 RESULTS AND DISCUSSION

The results of all our experiments on different instances of LSPE along with performance without using PE are presented in Table 1 whereas the comparison of the best results from Table 1 with baseline models and SOTA is shown in Table 2. We now summarize our observations and insights.

Table 1: Results on the ZINC, OGBG-MOLTOX21 and OGBG-MOLPCBA datasets. All scores are averaged over 4 runs with 4 different seeds. **Bold**: GNN's best score, **Red**: Dataset's best score.

| | Model | Init PE | LSPE | PosLoss | L | #Param | TestMAE±s.d. | TrainMAE±s.d. | Epochs | Epoch/Total |
|---|---|---|---|---|---|---|---|---|---|---|
| ZINC | GatedGCN | x | x | x | 16 | 504309 | 0.251±0.009 | 0.025±0.005 | 440.25 | 8.76s/1.08hr |
| | GatedGCN | LapPE | x | x | 16 | 505011 | 0.202±0.006 | 0.033±0.003 | 426.00 | 8.91s/1.22hr |
| | GatedGCN | RWPE | ✓ | x | 16 | 522870 | 0.093±0.003 | 0.014±0.003 | 440.75 | 15.17s/1.99hr |
| | GatedGCN | RWPE | ✓ | ✓ | 16 | 522870 | **0.090±0.001** | 0.013±0.004 | 460.50 | 33.06s/4.39hr |
| | PNA | x | x | x | 16 | 369235 | 0.141±0.004 | 0.020±0.003 | 451.25 | 79.67s/10.03hr |
| | PNA | RWPE | ✓ | x | 16 | 503061 | **0.095±0.002** | 0.022±0.002 | 462.25 | 127.69s/16.61hr |
| | SAN | x | x | x | 10 | 501314 | 0.181±0.004 | 0.017±0.004 | 433.50 | 74.33s/9.23hr |
| | SAN | RWPE | ✓ | x | 10 | 588066 | **0.104±0.004** | 0.016±0.002 | 462.50 | 134.74s/17.23hr |
| | GraphiT | x | x | x | 10 | 501313 | 0.181±0.006 | 0.021±0.003 | 493.25 | 63.54s/9.37hr |
| | GraphiT | RWPE | ✓ | x | 10 | 588065 | **0.106±0.002** | 0.028±0.002 | 420.50 | 125.39s/14.84hr |
| | **Model** | **Init PE** | **LSPE** | **PosLoss** | **L** | **#Param** | **TestAUC±s.d.** | **TrainAUC±s.d.** | **Epochs** | **Epoch/Total** |
| MOLTOX21 | GatedGCN | x | x | x | 8 | 1003739 | 0.772±0.006 | 0.933±0.010 | 304.25 | 5.12s/0.46hr |
| | GatedGCN | LapPE | x | x | 8 | 1004355 | 0.774±0.007 | 0.921±0.006 | 275.50 | 5.23s/0.48hr |
| | GatedGCN | RWPE | ✓ | x | 8 | 1063821 | **0.775±0.003** | 0.906±0.003 | 246.50 | 5.99s/0.63hr |
| | PNA | x | x | x | 8 | 5244849 | 0.755±0.008 | 0.876±0.014 | 214.75 | 6.25s/0.38hr |
| | PNA | RWPE | ✓ | x | 8 | 5357393 | **0.761±0.007** | 0.871±0.009 | 215.50 | 7.61s/0.56hr |
| | PNA | RWPE | ✓ | ✓ | 8 | 5357393 | **0.758±0.003** | 0.875±0.012 | 194.25 | 18.09s/1.07hr |
| | SAN | x | x | x | 10 | 957871 | **0.744±0.007** | 0.915±0.015 | 279.75 | 18.06s/1.44hr |
| | SAN | RWPE | ✓ | x | 10 | 1051017 | 0.744±0.008 | 0.918±0.018 | 281.75 | 30.82s/2.84hr |
| | GraphiT | x | x | x | 10 | 957870 | 0.743±0.003 | 0.919±0.023 | 276.50 | 16.73s/1.36hr |
| | GraphiT | RWPE | ✓ | x | 10 | 1051788 | **0.746±0.010** | 0.934±0.016 | 279.75 | 27.92s/2.57hr |
| | **Model** | **Init PE** | **LSPE** | **PosLoss** | **L** | **#Param** | **TestAP±s.d.** | **TrainAP±s.d.** | **Epochs** | **Epoch/Total** |
| MOLPCBA | GatedGCN | x | x | x | 8 | 1008263 | 0.262±0.001 | 0.401±0.057 | 190.50 | 149.10s/7.91hr |
| | GatedGCN | LapPE | x | x | 8 | 1008879 | 0.266±0.002 | 0.391±0.003 | 177.00 | 152.94s/8.29hr |
| | GatedGCN | RWPE | ✓ | x | 8 | 1068721 | **0.267±0.002** | 0.403±0.006 | 181.00 | 206.43s/11.64hr |
| | PNA | x | x | x | 4 | 6550839 | 0.279±0.003 | 0.448±0.004 | 129.25 | 174.75s/6.34hr |
| | PNA | RWPE | ✓ | x | 4 | 6521029 | **0.284±0.002** | 0.383±0.005 | 320.00 | 201.05s/22.99hr |

**No PE results in lowest performance.** In Table 1, the GNNs which do not use PE tend to give the worse performance on all the three datasets. This finding is aligned to the recent literature (Sec. B.2) that has guided research towards powerful PE methods for expressive GNNs. Besides, it can be observed that the extent of poor performance of models without PE against using a PE (LapPE or LSPE) is greater for ZINC than the two OGBG-MOL* datasets used. This difference can be explained by the fact that ZINC features are purely atom and bond descriptors whereas OGB-MOL* features consist additional information that is informative of e.g. if an atom is in ring, among others.

Table 2: Comparison of our best `LSPE` results from Table 1 with baselines and state-of-the-art GNNs (Sec. A.4) on each dataset. For ZINC, all the scores in Table 2a are the models with the ∼500k parameters. The scores on OGBG-MOL* in Tables 2b and 2c are taken from the OGB project and its leaderboards (Hu et al., 2020), where models have different number of parameters.

(a) ZINC

| Model | Test MAE |
|---|---|
| GCN | 0.367±0.011 |
| GAT | 0.384±0.007 |
| GatedGCN-LapPE | 0.202±0.006 |
| GT | 0.226±0.014 |
| SAN | 0.139±0.006 |
| Graphormer | 0.122±0.006 |
| GatedGCN-LSPE | **0.090±0.001** |

(b) OGBG-MOLTOX21

| Model | Test ROC-AUC |
|---|---|
| GCN | 0.7529±0.0069 |
| GCN-VN | 0.7746±0.0086 |
| GIN | 0.7491±0.0051 |
| GIN-VN | **0.7757±0.0062** |
| GatedGCN-LapPE | 0.7743±0.0073 |
| GatedGCN-LSPE | **0.7754±0.0032** |

(c) OGBG-MOLPCBA

| Model | Test AP |
|---|---|
| GIN | 0.2266±0.0028 |
| GIN-VN | 0.2703±0.0023 |
| DeeperGCN-VN | 0.2781±0.0038 |
| PNA | 0.2838±0.0035 |
| DGN | 0.2885±0.0030 |
| PHC-GNN | **0.2947±0.0026** |
| PNA-LSPE | 0.2840±0.0021 |

**`LSPE` boosts the capabilities of existing GNNs.** Both sparse GNNs and Transformer GNNs are improved significantly when they are augmented with `LSPE` having RWPE as initial PE, see Table 1. For instance, the best GNN without PE for ZINC, i.e. PNA, gives an improvement of 32.62% (0.095 vs. 0.141) when `LSPE` is used to learn the structural and positional representations in a decoupled manner. On other GNNs, this boost is even higher, see GatedGCN-LSPE which shows a gain of 64.14% (0.090 vs. 0.251). On MOLTOX21, PNA-LSPE improves 0.79% (0.761 vs. 0.755) over PNA while the remaining models show either minor gains or attain the same performance when not using PE. This consistent trend is also observed for MOLPCBA where `LSPE` boosts PNA by 1.79%.

**Sparse vs. Transformer GNNs.** When we compare the performance of sparse GNNs (GatedGCN, PNA) against Transformer GNNs (SAN, GraphiT) augmented with `LSPE` in Table 1, the performance of the sparse GNNs is surprisingly better than the latter, despite Transformer GNNs being theoretically well-posed to counter the limitations of long-range interactions of the former. Notably, the evaluation of our proposed architecture, in this work, is on molecular graphs on which the information among local structures seems to be the most critical, diminishes the need of full attention. This also aligns with the insight put forward in Kreuzer et al. (2021) where the SAN, a Transformer model, benefited less from full attention on molecules. Beyond molecular graphs, there may be other domains where Transformer GNNs could give better performance, but still these would not scale in view of the quadratic computational complexity. Indeed, it is important to notice the much lesser training times of sparse GNNs compared to Transformer GNNs in Table 1.

**`LSPE` improves the state-of-the-art for domain-agnostic GNNs.** When we compare the best performing instantiation of the `LSPE` from Table 1 with baseline GNN models from the literature on the three benchmark datasets, our proposed architecture improves the SOTA on ZINC, while achieving SOTA-comparable performance on remaining datasets, see Table 2. On ZINC, GatedGCN-LSPE surpasses most baselines by a large margin to give a test MAE of 0.090 which is an improvement of 35.25% and 26.23% respectively over the two recent-most Transformer based GNNs, SAN and Graphormer. On MOLTOX21, GatedGCN-LSPE reports a test ROC-AUC score of 0.7754 which is similar to the best baseline GIN (0.7757) that uses virtual node (VN). Finally, `LSPE` enables PNA to achieve comparable performance to SOTA on MOLPCBA while boosting its performance when no PE was used. We note here that ZINC scores can even be boosted beyond `LSPE`'s SOTA when domain expertise is used (Bouritsas et al., 2020; Bodnar et al., 2021) while Graphormer (Ying et al., 2021) achieved the top score on MOLPCBA when pre-trained on a very large (3.8M graphs) dataset. To ensure fair comparison with other scores, we did not use these two results in Table 2.

**On Positional loss.** It can be observed in Table 1 that the positional loss Eq. (12), further pushes the best `LSPE` score on ZINC slightly from 0.093 to 0.090, while on MOLTOX21 it only improves the train score though obtaining comparable test performance. We will investigate a more consistent positional loss in a future work.

Finally, we would like to highlight the generic nature of our proposed architecture which can be applied to any MP-GNN in practice as demonstrated by four diverse GNNs in this work.

## 4.3 ABLATION STUDIES

Through ablation studies, we show – i) the usefulness of learning positional representation at every layer vs. simply injecting a pre-computed positional encoding in the input layer, and ii) the selection of the number of $k$ for the steps in RWPE in the proposed `LSPE` architecture.

Table 3: Comparing the final `LSPE` architecture against simpler models which add pre-computed PE at input layer (or final layer) of a GNN, using GatedGCN model on ZINC. The column 'Final $h$' denotes whether only the node structural features are used as final node features (denoted by $h^L$), or are concatenated with (i) node positional features (denoted by $[h^L, p^L]$) at the final layer, (ii) pre-computed RWPE (denoted by $[h^L, \mathbf{RWPE}]$).

| Model | Init PE | LSPE | Final $h$ | $L$ | #Param | Test MAE±s.d. | Train MAE±s.d. | #Epochs | Epoch/Total |
|---|---|---|---|---|---|---|---|---|---|
| GatedGCN | x | x | $h^L$ | 16 | 504309 | 0.251±0.009 | 0.025±0.005 | 440.25 | 8.76s/1.08hr |
| GatedGCN | **LapPE** | x | $h^L$ | 16 | 505011 | 0.202±0.006 | 0.033±0.003 | 426.00 | 8.91s/1.22hr |
| GatedGCN | **RWPE** | x | $h^L$ | 16 | 505947 | 0.122±0.003 | 0.013±0.003 | 436.25 | 9.14s/1.28hr |
| GatedGCN | x | x | $[h^L, \mathbf{RWPE}]$ | 16 | 515249 | 0.249±0.012 | 0.024±0.002 | 437.50 | 10.05s/1.55hr |
| GatedGCN | **LapPE** | ✓ | $h^L$ | 16 | 516722 | 0.202±0.008 | 0.032±0.005 | 405.25 | 15.10s/1.84hr |
| GatedGCN | **LapPE** | ✓ | $[h^L, p^L]$ | 16 | 520734 | 0.196±0.008 | 0.023±0.004 | 454.00 | 15.22s/2.06hr |
| GatedGCN | **RWPE** | ✓ | $h^L$ | 16 | 518150 | 0.100±0.006 | 0.018±0.012 | 395.00 | 15.09s/1.73hr |
| GatedGCN | **RWPE** | ✓ | $[h^L, p^L]$ | 16 | 522870 | 0.093±0.003 | 0.014±0.003 | 440.75 | 15.17s/1.99hr |

**Learning PE at every layer provides the best performance.** In Table 3, GatedGCN-RWPE corresponds to the model where LapPE are replaced with $k$-dim pre-computed random walk features at the first layer, and the PE are not updated in the subsequent layers. First, we observe a significant leap in performance (from $0.202$ to $0.122$) when the RWPE are injected in place of LapPE at the first layer, suggesting that RWPE could encode better positional information in GNNs as they are not limited by the sign ambiguity of LapPE. See Section A.1 in the supplementary material for an example of RWPE's representation power. Note that the injection of RWPE at the final layer instead of the input layer gives poor performance. The reason behind the better performance of concatenating RWPE at the input layer is to inform the GNN aggregation function of the node positions in order to distinguish them in the case of graph symmetries like isomorphic nodes.

Now, if we observe the training performance, GatedGCN-RWPE leads to an overfit on ZINC. However, when the positional representations are also updated, the overfit is considerably alleviated improving the test score to $0.100$. Finally, when we further fuse the learned positional features at the final layer with the structural features, Eq. (12), the model achieves the best MAE test of $0.093$. This study justifies how the GNN model learns best when the positional representations can be tuned and better adjusted to the learning task being dealt with.

**The choice of $k$ steps to initialize RWPE.** In Figure 7 (see Section A.5), we study the effect of choosing a suitable number of $k$ steps for the random walk features that are used as initial positional encoding in Section 3.1. This value $k$ is also used to set the final dimension of the learned positional representation in the last layer. Numerical experiments show the best values of $k$ to be $20$ and $16$ for ZINC with GatedGCN-LSPE and OGBG-MOLTOX21 with PNA-LSPE respectively, which are larger values from what was used in Li et al. (2020b) ($k = 3, 4$) where the RW features are treated as distance encoding. The difference of $k$ value is due to two reasons. First, the proposed RWPE requires to use a large $k$ value to possibly provide a unique node representation with different $k$-hop neighborhoods. Second, Li et al. (2020b) not only uses $RW_{ii}^k$ but also considers all pairwise $RW_{ij}^k$ between nodes $i$ and $j$ in a target set of nodes, which increases the computational complexity.

## 5 CONCLUSION

This work presents a novel approach to learn structural and positional representations separately in a graph neural network. The resultant architecture, `LSPE` enables a principled and effective learning of these two key properties that make GNN representation even more expressive. Main design components of `LSPE` are – i) higher-order position informative random walk features as PE initialization, ii) decoupling positional representations at every GNN layer, and iii) the fusion of the structural and positional features finally to generate hybrid features for the learning task. We observe a consistent increase of performance across several instances of our model on the benchmark datasets used for evaluation. Our architecture is simple and universal to be used with any sparse GNNs or Transformer GNNs as demonstrated by two sparse GNNs and two fully connected Transformer based GNNs in our numerical experiments. Given the importance of incorporating expressive positional encodings to theoretically improve GNNs as seen in the recent literature, we believe this paper provides a useful architectural framework that can be considered when developing future models which improve graph positional encodings, for both GNNs and Transformers.

ETHICS STATEMENT

In this work, we present an approach to improve neural network methods for graphs by considering efficient learnable positional encoding while keeping the linear complexity of the model w.r.t to the number of nodes. This improves the cost of training such models, as contrast to some previous works that improved GNNs at the cost of higher-order tensor computation. We discover another insight that the linear complexity models (sparse GNNs) can outperform quadratic complexity models (Transformers). Consequently, one beneficial impact of our work is that its use can reduce GPU and computational resources, eventually contributing to minimizing the adverse effect of deep learning training on environment. However, the method we propose belongs to a class of architectures that can be used on malicious applications since the internet and several of the processes in its ecosystem can be represented in form of graphs. To prevent such applications, ethical guidelines can be set and enforced which constraint the usage of our proposed model.

REPRODUCIBILITY STATEMENT

The authors support and advocate the principles of open science and reproducible research. The algorithms and architectures proposed in this work are open-sourced in a free and public code repository with easy-to-use scripts to reproduce different experiments and evaluations presented. The tables included in the paper mention critical details on the number of layers and the total number of model parameters that are trained. Similarly, the visualization and illustrations presented in the main paper as well as the supplementary material contain the exact details on the dataset examples (such as index) used. Finally, a detailed table consisting of several hyperparameters used for the experiments are included in the supplementary, ensuring the reproducibility of the results discussed in this work.

ACKNOWLEDGMENTS

XB is supported by NRF Fellowship NRFF2017-10 and NUS-R-252-000-B97-133. This research is supported by Nanyang Technological University, under SUG Grant (020724-00001). VPD would like to thank Andreea Deac for her helpful feedback, Quan Gan for his support on the DGL library, Gabriele Corso for answering questions related to the PNA model, and Chaitanya K. Joshi for useful comments. Finally, the authors would like to thank the anonymous reviewers for their helpful suggestions and feedbacks.

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

## A    SUPPLEMENTARY

### A.1    DISTINGUISHING NON-ISOMORPHIC GRAPHS USING RANDOM WALK FEATURES

The choice of the initial PE in our proposed architecture can be several based on graph diffusion or other related techniques. In this section, we study RWPE (Eqn. 10) which we initialize with $k$-steps of random walk. Precisely we use a $k$-dim vector that encodes the landing probabilities of a node $i$ to itself in $1$ to $k$ steps. This initial PE vector for a node $i$ is given by $[\text{RW}_{ii}, \text{RW}_{ii}^2, \ldots, \text{RW}_{ii}^k] \in \mathbb{R}^k$ which is pre-computed before the model training. Here, we demonstrate that such PE vector can help distinguish i) structurally dissimilar nodes and ii) non-isomorphic graphs on which 1-WL, and equivalently MP-GNNs, fail, thus illustrating the empirically powerful nature of MPGNNs-LSPE that relies on this choice of positional features initialization.

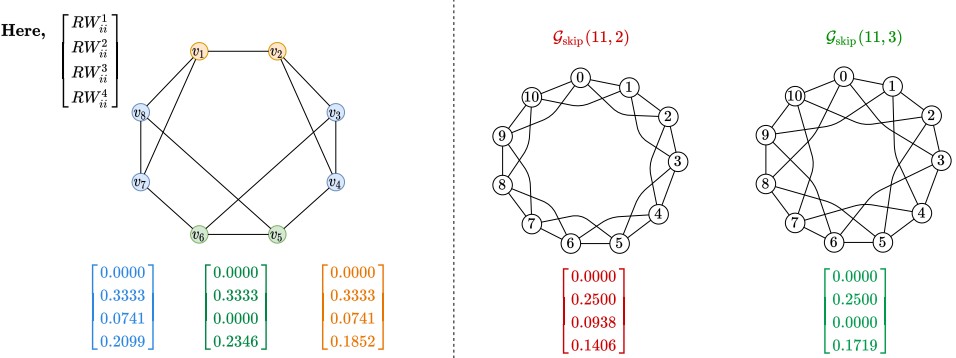

Figure 3: **Left:** Example 3-regular graph with 8 nodes from Li et al. (2020b) where the nodes are structurally different and colored accordingly. The $4$-dim initial RWPE vector is shown against the corresponding nodes with their respective colors. **Right:** Example pair of non-isomorphic graphs with 11 nodes and skip-links 2 and 3 from Murphy et al. (2019). Each node in a graph gets the same $4$-dim RWPE vector, and shown above in colors are the respective graphs' RWPE vectors after averaging across all the nodes.

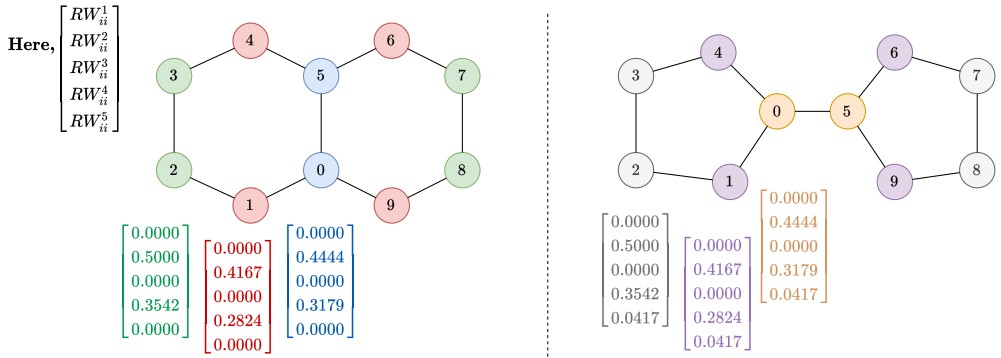

Figure 4: A pair of non-isomorphic and non-regular graphs (Left: Decalin, Right: Bicyclopentyl) from Sato (2020). The $5$-dim initial PE vector is shown against the corresponding nodes with their respective colors.

We show the simulation of the nodes' initial RWPE vectors on three examples in Figure 3 (Left), Figure 3 (Right), and Figure 4 where the graphs either do not have any node attributes (Figure 3), or have the same node attributes (Figure 4 where each node denotes a Carbon atom). When we apply MPGNNs on the graph in 3 (Left), each node will have the same feature representation as it is a regular graph without any node attributes. However, there are structurally 3 different kinds of nodes denoted by the same number of different colors. If we initialize the PE for these nodes for $k = 4$

random walk steps, we can observe that the nodes are being assigned the $4$-dim feature vectors that is consistent to their initial structural roles in the graph, thus being distinguishable.

Similarly, Figure 3 (Right) is a pair of non-isomorphic graphs from the theoretically challenging and highly symmetric Circulant Skip Link (CSL) dataset from Murphy et al. (2019). It can be noticed that every node in a graph here has the same structural role as the each node has edges with other nodes at same hops. However, in $\mathcal{G}_{\text{skip}}(11, 2)$, the edges are between nodes at $1, 2$ hops whereas in $\mathcal{G}_{\text{skip}}(11, 3)$, the edges are between the nodes at $1, 3$ hops, with $2$ and $3$ being the skip-links of the two graphs, respectively. In such a scenario, the node in the $\mathcal{G}_{\text{skip}}(11, 2)$ gets a different $4$-dim initial PE than a node in $\mathcal{G}_{\text{skip}}(11, 3)$, thus helping eventually to distinguish the two graphs when these node features are pooled to generate the graph feature vector.

Finally, in Figure 4, a pair of non-isomorphic and non-regular graphs is shown from Sato (2020) that MPGNNs fail to distinguish. If we use $5$ steps of Random Walk to initialize the node's PE vector, we can observe that the two graphs can easily be distinguished. We note here that the random walk based PE initialization (RWPE) is close to one of the Distance Encoding instantiations used in Li et al. (2020b). However, we do not require to consider pairwise scores $\text{RW}_{ij}^k$ between nodes $i$ and $j$ and any sub-set of nodes from the original graph, thus making our method less computationally demanding.

## A.2 RANDOM WALK PE FEATURE AND GRAPH ISOMORPHISM TEST

Similar to the 1-WL test for graph isomorphism (Weisfeiler & Leman, 1968; Morris et al., 2019; Sato, 2020), the RWPE can be used as a node coloring algorithm to test if two graphs are non-isomorphic, as described in Algorithm 1. Note that this algorithm cannot guarantee that two graphs are isomorphic, like the WL test. However, our analysis in Section A.1 shows this algorithm to be strictly powerful than the 1-WL test as the pairs of graphs in Figure 3 (Right) and in Figure 4 are not distinguishable by 1-WL. Although this increase in power is being achieved without the need of maintaining colors for tuple of nodes to encode higher order interactions (as in k-WL), the algorithm's complexity is of $O(k * n^3)$ due to the matrix multiplication in Step 5 (b) and Step 5 (c), compared to $O(k * n^2)$ of 1-WL, with $k$ being the number of iterations until convergence.

---

**Algorithm 1** Algorithm to decide whether a pair of graphs are not isomorphic based on random walk landing probabilities of each node to itself.

---

**Input:** A pair of graphs $\mathcal{G}_1 = (\mathcal{V}_1, \mathcal{E}_1)$, $\mathcal{G}_2 = (\mathcal{V}_2, \mathcal{E}_2)$ with $n$ nodes and $e$ edges in each graph. $A_1 \in \mathbb{R}^{n \times n}$ and $A_2 \in \mathbb{R}^{n \times n}$ denote the adjacency matrices, $D_1 \in \mathbb{R}^{n \times n}$ and $D_2 \in \mathbb{R}^{n \times n}$ denote the degree matrices of graphs $\mathcal{G}_1$ and $\mathcal{G}_2$ respectively.
**Output:** Return "non-isomorphic" if $\mathcal{G}_1$ and $\mathcal{G}_2$ are not isomorphic else "possibly isomorphic".

1. $M^{(0)} \leftarrow A_1 D_1^{-1} \in \mathbb{R}^{n \times n}$

2. $N^{(0)} \leftarrow A_2 D_2^{-1} \in \mathbb{R}^{n \times n}$

3. $c_u^{(0)} \leftarrow M_{u,u}^{(0)} \quad \forall u \in \mathcal{V}_1$

4. $d_v^{(0)} \leftarrow N_{v,v}^{(0)} \quad \forall v \in \mathcal{V}_2$

5. for $k = 1, 2, \cdots$ (until convergence to stationary distribution)

   (a) if $\text{HASH}\Big(\{\{c_u^{(k-1)} \in \mathbb{R}^k \mid u \in \mathcal{V}_1\}\}\Big) \neq \text{HASH}\Big(\{\{d_v^{(k-1)} \in \mathbb{R}^k \mid v \in \mathcal{V}_2\}\}\Big)$ then return "non-isomorphic"

   (b) $M^{(k)} \leftarrow M^{(k-1)} M^{(0)} \in \mathbb{R}^{n \times n}$

   (c) $N^{(k)} \leftarrow N^{(k-1)} N^{(0)} \in \mathbb{R}^{n \times n}$

   (d) $c_u^{(k)} \leftarrow$ append $M_{u,u}^{(k)}$ to $c_u^{(k-1)} \quad \forall u \in \mathcal{V}_1$

   (e) $d_v^{(k)} \leftarrow$ append $N_{v,v}^{(k)}$ to $d_v^{(k-1)} \quad \forall v \in \mathcal{V}_2$

6. return "possibly isomorphic"

---

where $\text{HASH}$ is an injective hash function and $\{\{\dots\}\}$ denotes a multiset.

## A.3 STUDY OF LAPPE AND RWPE AS INITIAL PE

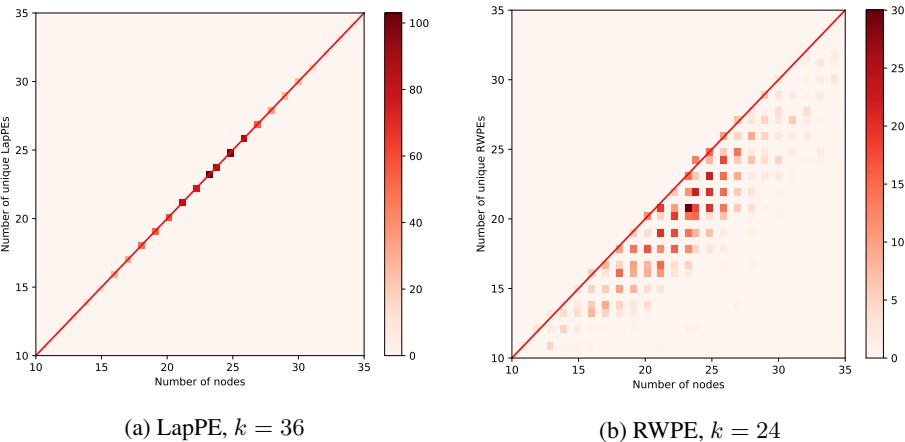

(a) LapPE, $k = 36$

(b) RWPE, $k = 24$

Figure 5: Plot of the number of nodes in a graph vs. the number of unique PE for LapPE and RWPE. A point in the plots represents a graph in the ZINC validation set (composed of 1000 graphs) where the $x$-axis is the number of nodes, the $y$-axis is the number of unique PEs and the point intensity is the number of graphs with the same pair $(x, y)$. Besides, Fig. 5a has 36-dim LapPE (trailing dims padded with zero for a graph with $n < 36$), and Fig. 5b has 24-dim RWPE.

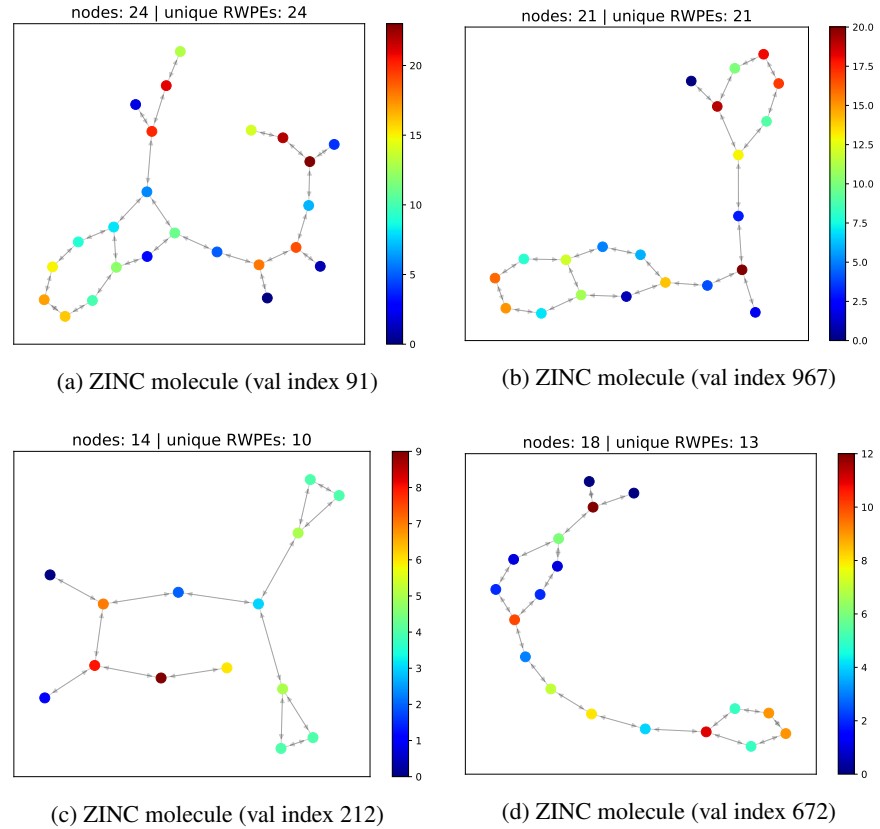

(a) ZINC molecule (val index 91)

(b) ZINC molecule (val index 967)

(c) ZINC molecule (val index 212)

(d) ZINC molecule (val index 672)

Figure 6: Sample graph plots from the ZINC validation set with each node color in a graph representing a unique RWPE vector, when $k = 24$. The corresponding graph ids, the number of nodes in the graphs and the number of unique RWPEs are labelled against the figures.

Figure 5 visualizes the uniqueness of the node representation with LapPE and RWPE (which serve as initial PE of our network) using the ZINC validation set of 1000 real-world molecular graphs. If the

initial PE is unique for each node in a graph, then the graph lies on the straight diagonal line. Figure 5a shows the result for LapPE, all graphs lie on the diagonal line as Laplacian eigenvectors guarantee unique node coordinates in the Euclidean transformed space. Figure 5b presents the result for RWPE. We observe that not all, but a large amount of ZINC molecular graphs stay close to the straight line, showing that most graphs have a large amount of nodes with unique RWPE. For example, there are 30 graphs with 24 nodes having 21 unique RWPE, equivalent to 87.5% of nodes with unique PE.

Additionally, we visualize four sample graph plots from the ZINC validation set in Figure 6 where the first two graphs have completely unique RWPE features, while the next two graphs have partially unique RWPEs (71.43% and 72.22% respectively). The visualization assigns a unique node color for each unique RWPE representation. Therefore, graphs in Figures 6a and 6b are plotted with each node assigned to a unique color based on their RWPE features, and graphs in Figures 6c and 6d are represented with 10 and 13 unique colors respectively corresponding to their number of unique RWPE representations. In particular, observe the green-shade colored nodes in Figure 6c (top and bottom-right) as well as blue-shade (mid-left) and orange-shade (bottom-right) colored nodes in Figure 6d. We can easily see that the nodes with the same color are isomorphic in the graph, i.e. their $k$-hop structural neighborhoods are the same for values $k \geq 11$.

We remind that RWPE provides a unique node representation under the condition that each node have a unique $k$-hop topological neighborhood for a sufficient large $k$. While this condition is experimentally true for most nodes, it is not always satisfied. But despite this approximation, for a sufficiently large number $k$ of random walk iterations, RWPE is still able to capture global higher-order positioning of nodes that is used as initial PE, and is beneficial to the proposed LSPE architecture as demonstrated by the gain of performance in several experiments.

### A.4 MODELS USED FOR COMPARISON IN TABLE 2

As a complete reference, the different GNN baselines and SOTA models that are used for the comparison in Table 2 are Graph Convolutional Networks (GCN) (Kipf & Welling, 2017), Graph Attention Networks (GAT) (Veličković et al., 2018), GatedGCN-LapPE (Bresson & Laurent, 2017; Dwivedi et al., 2020), Graph Transformer (GT) (Dwivedi & Bresson, 2021), Spectral Attention Networks (SAN) (Kreuzer et al., 2021), Graphormer (Ying et al., 2021), Graph Isomorphism Networks (GIN) (Xu et al., 2019), DeeperGCN (Li et al., 2020a), Principle Neighborhood Aggregation (PNA) (Corso et al., 2020), Directional Graph Networks (DGN) (Beani et al., 2021) and Parameterized Hypercomplex GNNs (PHC-GNN) (Le et al., 2021).

### A.5 FIGURE FOR THE STUDY OF $k$ STEPS IN RWPE (SECTION 4.3)

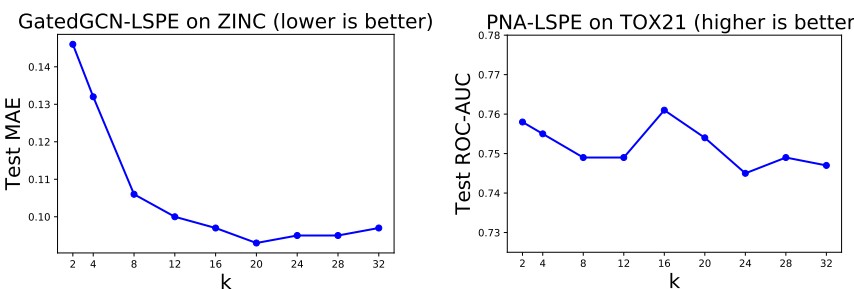

Figure 7: Test scores on selecting different values of $k$ which is used to determine the number of iterative steps of RW in RWPE as well as the dimension of the learned PE at the final layer, Eqn. 12.

## B RELATED WORK IN DETAIL

In this detailed section on related work, we first review the limitations of existing MP-GNN architectures in terms of their theoretical expressiveness, suggesting possible improvements to make GNNs more powerful. Then, we introduce a number of non-learned and learning techniques that can be studied under the umbrella of graph positional encoding. Finally, we highlight the recent developments for generalizing Transformers to graphs. Our aim is to connect meaningful innovations through

the detailed background on these three research directions, the unification of which spearheaded the development of this work.

## B.1 THEORETICAL EXPRESSIVITY AND WEISFEILER-LEMAN GNNS

**Weisfeiler-Leman test.** The limitation of MP-GNNs in failing to distinguish non-isomorphic graphs was first carefully studied in Xu et al. (2019) and Morris et al. (2019), based on the equivalence of MP-GNNs and the 1-WL isomorphism test (Weisfeiler & Leman, 1968). As such, MP-GNNs may perform poorly on graphs that exhibit several symmetries in their original structure, such as node and edge isomorphisms (Murphy et al., 2019; Srinivasan & Ribeiro, 2019). Besides, some message-passing functions may not be discriminative enough (Xu et al., 2019; Corso et al., 2020).

**Equivariant GNNs.** Graph Isomorphism Networks (GINs) (Xu et al., 2019) were designed to be as maximally expressive as the original 1-WL test (Weisfeiler & Leman, 1968). However, the 1-WL test can fail to distinguish (simple) non-isomorphic graphs, thus requiring novel GNNs with more expressivity power. As the original 1-WL test only considers 2-tuple of nodes, i.e. the standard edges in a graph, a natural approach to improve the expressivity power of the 1-WL test is to examine higher-order interactions between nodes with $k$-tuple of nodes with $k \geq 3$. To this end, $k$-order Equivariant-GNNs were introduced in Maron et al. (2018). But these architectures require $O(n^k)$ memory and speed complexities. This is an important practical limitation as $k = 3$ is at least needed to design more powerful GNNs than GINs. Along this line, the most efficient WL-GNNs that have been proposed are in Maron et al. (2019); Chen et al. (2019); Azizian & Lelarge (2020), which have $O(n^2)$ memory and $O(n^3)$ speed complexities.

## B.2 GRAPH POSITIONAL ENCODING

**Importance of Positional Information.** The idea of positional encoding, i.e. the notion of global position of pixels in images, words in texts and nodes in graphs, plays a central role in the effectiveness of the most prominent neural networks with ConvNets (LeCun et al., 1998), RNNs (Hochreiter & Schmidhuber, 1997), and Transformers (Vaswani et al., 2017). These architectures integrate structural and positional attributes of data when building abstract feature representations. For instances, ConvNets intrinsically consider regular spatial structure for the position of pixels (Islam et al., 2020), RNNs also build on the sequential structure of the word positions, and Transformers employ positional encoding of words (see Dufter et al. (2021) for a review). For GNNs, the position of nodes is more challenging due to the fact that there does not exist a canonical positioning of nodes in arbitrary graphs. This implies that there is no obvious notion of global and relative position of nodes, and consequently no specific directions on graphs (like the top, down, left and right directions in images). Despite these issues, graph positional encoding are as much critical for GNNs as they are for ConvNets, RNNs and Transformers, as demonstrated for prediction tasks on graphs (Srinivasan & Ribeiro, 2019; Cui et al., 2021).

**Index Positional Encoding.** Loukas (2020) identified another cause of the limited expressivity of the standard MP-GNNs. Such GNNs do not have the capacity to handle anonymous nodes, i.e. nodes which do not have unique node features. This property turns out to be critical to show that MP-GNNs can be universal approximators if each node in the graph can be assigned to a unique or discriminating feature. The theorem results from an alignment between MP-GNNs and distributed local algorithms (Naor & Stockmeyer, 1995; Sato et al., 2019). In order to address the issue of anonymous MP-GNNs and improve their theoretical expressiveness w.r.t the WL test, Murphy et al. (2019) introduced Graph Relational Pooling. Their model assigns a unique identifier to each node, defined by an indexing of the nodes. However, such a model must be trained with the $n!$ possible index permutations to guarantee higher expressivity, which is not computationally feasible. As a consequence, during training, node indexing is uniformly sampled from the $n!$ possible choices in order for their network to learn to be independent to the choice of the index PE at test time. Similarly, random node identifier could be used for breaking the node anonymity. Yet, this PE also suffers from the lack of generalization for unseen graphs (Loukas, 2020).

**Laplacian Positional Encoding.** Besides providing a unique representation for each node, meaningful graph PE should also be permutation-invariant and distance-sensitive, meaning that the difference between the PEs of two nodes far apart on the graph must be large, and small for two nodes nearby. Laplacian eigenvectors (Belkin & Niyogi, 2003) appear to be good candidates for graph PE, belonging

to the class of unsupervised manifold learning techniques. Precisely, they are spectral techniques that embed graphs into an Euclidean space, and are defined via the factorization of the graph Laplacian $\Delta = I_n - D^{-1/2}AD^{-1/2} = U^T\Lambda U$, where $I_n$ is the $n \times n$ identity matrix, $A$ the $n \times n$ adjacency matrix, $D$ the $n \times n$ degree matrix, and $n \times n$ matrices $\Lambda$ and $U$ correspond to the eigenvalues and eigenvectors respectively. The complexity for computing this full factorization is $O(E^{3/2})$ and $O(n)$ with approximate Nystrom method (Fowlkes et al., 2004). Laplacian eigenvectors form a meaningful local coordinate system, while preserving the global graph structure. As these eigenvectors hold the key properties of permutation-invariant, uniqueness, computational efficiency and distance-aware w.r.t. the graph topology, they were proposed as graph PE (Dwivedi et al., 2020; Dwivedi & Bresson, 2021). They also naturally generalize the positional encoding used in Transformers (Vaswani et al., 2017) to arbitrary graphs. The main limitation of this graph PE is the existence of a sign ambiguity as eigenvectors are defined up to $\pm 1$. This leads to $2^k$ number of possible sign values when selecting $k$ number of eigenvectors. In practice, we choose $k \leq n$ eigenvectors given the manifold assumption, and therefore $2^k$ is much smaller $n!$ (the number of possible ordering of the nodes), and therefore smaller amount of ambiguities to be resolved by the network. During the training, eigenvectors are uniformly sampled at random between the $2^k$ possibilities (Dwivedi et al., 2020; Kreuzer et al., 2021) in order for the network to learn to be invariant w.r.t the sign of the eigenvectors.

**Other graph PE.** Li et al. (2020b) proposed the use of distance encoding (DE) as node attributes, and additionally as controller of message aggregation. DE captures relative distances between nodes in a graph using powers of the random walk matrix. The resulting GNN was shown to have better expressivity than the 1-WL test. However, the limitation on regular graphs, and the cost and memory requirement of using power matrices may prevent the use of this technique to larger graphs. Khasahmadi et al. (2020) used random walk with restart (Pan et al., 2004) as topological embeddings with the initial node features.

You et al. (2019) proposed learnable position-aware embeddings based on random anchor sets of nodes for pairwise nodes (or link) tasks. This work also seeks to develop positional encoding that can be learned along with the structural representation within the GNN. However, the random selection of anchors has its limitations, which makes their approach less generalizable on inductive tasks.

Bouritsas et al. (2020); Bodnar et al. (2021) introduced hybrid GNNs based on the WL-test and the message-passing aggregation mechanism. These networks use prior knowledge about a class of graphs of interest such as rings for molecules and cliques for social networks. The prior information is then encoded into MP-GNNs to obtain more expressive models by showing that the such GNNs are not less powerful than the 3-WL test. They obtained top performance on molecular datasets but the prior information regarding graph sub-structures needs to be pre-computed, and sub-graph matching and counting require $O(n^k)$ for $k$-tuple sub-structure. Besides, complexity of the message passing process depends linearly w.r.t. the size of the sub-graph structure. Note that the core idea of substructure counting with e.g. the number of rings associated to an atom provides a powerful higher-order structural information to the network and can improve significantly the tasks related to substructure counting.

### B.3 TRANSFORMER-BASED GNNS

MP-GNNs are GNNs that leverage the sparse graph structure as computational graph, allowing training and inference with linear complexity and making them scalable to medium and large-scale graphs. However, besides their low expressivity, these GNNs hold two important and well-identified limitations. Firstly, MP-GNNs are susceptible to the information bottleneck limitation a.k.a. over-squashing (Alon & Yahav, 2020) when messages from across distant nodes are aggregated to a node. Secondly, long-range interactions between far away nodes can also be limited, and require multiple layers that can suffer from the vanishing gradient problem. These limitations are similar to the ones present in Recurrent Neural Networks (RNNs) (Hochreiter & Schmidhuber, 1997), and can lead MP-GNNs to perform poorly on tasks where long-range interactions are necessary.

To overcome these limitations, it seems natural to use Transformer networks (Vaswani et al., 2017) which alleviates the long-range issue as 'everything is connected to everything'. However, it was found that the direct adoption of full-graph operable Transformers perform poorly compared to MP-GNNs on graph structured datasets (Dwivedi & Bresson, 2021). Besides, Transformer-based GNNs require to replace $O(n)$ complexity with $O(n^2)$. So these GNNs are limited to small graphs

like molecules and cannot scale to larger ones like social graphs or knowledge graphs. Dwivedi & Bresson (2021) designed a sparsely-connected architecture called GraphTransformer that reduces the complexity to $O(E)$ by considering the graph topology instead of connecting each node to all other nodes, similar to GATs (Veličković et al., 2018). Still, the GraphTransformer was unable to outperform SOTA GNNs on benchmark datasets. Along this line, Kreuzer et al. (2021) recently proposed Spectral Attention Networks (SANs), a fully-graph operable Transformer model that improves GraphTransformer (Dwivedi & Bresson, 2021) with two contributions. First, the authors designed a learnable PE module based on self-attention applied to the Laplacian eigenvectors, and injected this resultant PE into the input layer of the network. Second, SANs separated the parameters for real edges and complementary (non-real) edges, enabling the model to process the available sparse graph structure and long-range node connections in a learnable manner. However, their learned PE, based on the Laplacian eigenvectors, inherently exhibits the limitation of sign ambiguity. Kreuzer et al. (2021) attempted at alleviating the sign ambiguity through another architecture named Edge-Wise LPE. However, the architecture's complexity being $O(n^4)$ makes it a practically infeasible model. GraphiT (Mialon et al., 2021) and Graphormer (Ying et al., 2021) were also very recently developed as full-graph operable Transformers for graphs with the idea to weigh (or, control) the attention mechanism based on the graph topology. Specifically, GraphiT employs diffusion geometry to capture short-range and long-range graph information, and Graphormer uses shortest paths. Altogether, these works exploit different relative positional encoding information to improve the expressivity of Transformers for graphs.

## C  INSTANCES OF LSPE WITH SPARSE AND TRANSFORMER GNNS

### C.1  SPARSE GNNS WITH LSPE

In this section, we augment two MP-GNN architectures with learnable positional representation, namely GatedGCN (Bresson & Laurent, 2017) and PNA (Corso et al., 2020).

### C.1.1  GATEDGCN-LSPE

GatedGCNs (Bresson & Laurent, 2017) are sparse MP-GNNs equipped with a soft-attention mechanism that is able to learn adaptive edge gates to improve the message aggregation step of GCN networks (Kipf & Welling, 2017). We augment this model to develop GatedGCN-LSPE, defined as:

$$h^{\ell+1}, e^{\ell+1}, p^{\ell+1} = \text{GatedGCN-LSPE}\Big(h^\ell, e^\ell, p^\ell\Big),\; h \in \mathbb{R}^{n \times d}, e \in \mathbb{R}^{E \times d}, p \in \mathbb{R}^{n \times d}, \quad (17)$$

$$\text{with } h_i^{\ell+1} = h_i^\ell + \text{ReLU}\Big(\text{BN}\Big(A_1^\ell \begin{bmatrix} h_i^\ell \\ p_i^\ell \end{bmatrix} + \sum_{j \in \mathcal{N}(i)} \eta_{ij}^\ell \odot A_2^\ell \begin{bmatrix} h_j^\ell \\ p_j^\ell \end{bmatrix}\Big)\Big), \quad (18)$$

$$e_{ij}^{\ell+1} = e_{ij}^\ell + \text{ReLU}\big(\text{BN}\big(\hat{\eta}_{ij}^\ell\big)\big), \quad (19)$$

$$p_i^{\ell+1} = p_i^\ell + \tanh\Big(C_1^\ell p_i^\ell + \sum_{j \in \mathcal{N}(i)} \eta_{ij}^\ell \odot C_2^\ell p_j^\ell\Big), \quad (20)$$

$$\text{and } \eta_{ij}^\ell = \frac{\sigma\big(\hat{\eta}_{ij}^\ell\big)}{\sum_{j' \in \mathcal{N}(i)} \sigma\big(\hat{\eta}_{ij'}^\ell\big) + \epsilon}, \quad (21)$$

$$\hat{\eta}_{ij}^\ell = B_1^\ell h_i^\ell + B_2^\ell h_j^\ell + B_3^\ell e_{ij}^\ell, \quad (22)$$

where $h_i^\ell, e_{ij}^\ell, p_i^\ell, \eta_{ij}^\ell, \hat{\eta}_{ij}^\ell \in \mathbb{R}^d$, $A_1^\ell, A_2^\ell \in \mathbb{R}^{d \times 2d}$ and $B_1^\ell, B_2^\ell, B_3^\ell, C_1^\ell, C_2^\ell \in \mathbb{R}^{d \times d}$.

### C.1.2  PNA-LSPE

PNA (Corso et al., 2020) is a sparse MP-GNN model which uses a combination of node aggregators to overcome the theoretical limitation of a single aggregator. We propose PNA-LSPE whose layer

update equation is defined as:

$$h^{\ell+1}, p^{\ell+1} = \text{PNA-LSPE}\Big(h^\ell, e^0, p^\ell\Big), \ h \in \mathbb{R}^{n \times d}, e^0 \in \mathbb{R}^{E \times d}, p \in \mathbb{R}^{n \times d}, \tag{23}$$

$$\text{with} \ \ h_i^{\ell+1} = h_i^\ell + \text{LReLU}\Big(\text{BN}\Big(U_h^\ell\Big(\begin{bmatrix} h_i^\ell \\ p_i^\ell \end{bmatrix}, \bigoplus_{j \in \mathcal{N}(i)} M_h^\ell\Big(\begin{bmatrix} h_i^\ell \\ p_i^\ell \end{bmatrix}, e_{ij}^0, \begin{bmatrix} h_j^\ell \\ p_j^\ell \end{bmatrix}\Big)\Big)\Big)\Big), \tag{24}$$

$$p_i^{\ell+1} = p_i^\ell + \tanh\Big(U_p^\ell\Big(p_i^\ell, \bigoplus_{j \in \mathcal{N}(i)} M_p^\ell\Big(p_i^\ell, e_{ij}^0, p_j^\ell\Big)\Big)\Big), \tag{25}$$

$$\text{and} \ \ \bigoplus = \begin{bmatrix} I \\ S(D, \alpha=1) \\ S(D, \alpha=-1) \end{bmatrix} \otimes \begin{bmatrix} \mu \\ \sigma \\ \max \\ \min \end{bmatrix}, \tag{26}$$

where $\bigoplus$ is the principal aggregator designed in (Corso et al., 2020), LReLU stands for LeakyReLU activation, amd $U_h^\ell, U_p^\ell, M_h^\ell$ and $M_p^\ell$ are linear layers (or multi-layer perceptrons) with learnable parameters.

## C.2 TRANSFORMER GNNS WITH LSPE

The recently developed SAN (Kreuzer et al., 2021), GraphiT (Mialon et al., 2021) and Graphormer (Ying et al., 2021) are promising full-graph operable Transformers incorporating several methods to encode positional and structural features into the network. In the next sections, we expand these Transformer-based networks with the proposed LSPE architecture.

### C.2.1 SAN-LSPE

Like Transformers, Spectral Attention Networks (SAN) (Kreuzer et al., 2021) operate on full graphs although the network separates the parameters coming from existing edges and non-existing edges in the graph. Furthemore, the contribution of attentions from existing and non-existing edges are weighted by an additive positive scalar $\gamma$, which can be tuned for different tasks. SAN also considers a Learnable Positional Encoding (LPE) module which takes in Laplacian eigenvectors and transforms them into a fixed size PE with a self-attention encoder. This PE is then used in the main architecture in a manner similar to MP-GNNs-PE as defined in Eq. (5). We propose to extend SAN by replacing the LPE module with the LSPE architecture proposed in Section 3.1 where positional representation is learned in line with structural embedding at each GNN layer:

$$h^{\ell+1}, p^{\ell+1} = \text{SAN-LSPE}\Big(h^\ell, e^0, p^\ell\Big), \ h \in \mathbb{R}^{n \times d}, e^0 \in \mathbb{R}^{n \times n \times d}, p \in \mathbb{R}^{n \times d}, \tag{27}$$

$$\text{with} \ \ h_i^{\ell+1} = \text{BN}\big(\bar{h}_i^{\ell+1} + W_2^\ell \, \text{ReLU}\big(W_1^\ell \, \bar{h}_i^{\ell+1}\big)\big) \in \mathbb{R}^d \tag{28}$$

$$\bar{h}_i^{\ell+1} = \text{BN}\Big(h_i^\ell + \hat{h}_i^{\ell+1}\Big) \in \mathbb{R}^d, \tag{29}$$

$$\hat{h}_i^{\ell+1} = O^\ell\Big(\, \big\|_{k=1}^{H} \sum_{j \in \mathcal{V}} \frac{w_{ij}^{k,\ell}}{\sum_{j' \in \mathcal{V}} w_{ij'}^{k,\ell}} \, v_j^{k,\ell}\Big) \in \mathbb{R}^d, \tag{30}$$

$$w_{ij}^{k,\ell} = \begin{cases} \frac{1}{1+\gamma} \cdot \exp(A_{ij}^{k,\ell}) & \text{if } ij \in E \\ \frac{\gamma}{1+\gamma} \cdot \exp(\bar{A}_{ij}^{k,\ell}) & \text{if } ij \notin E \end{cases}, \tag{31}$$

$$\begin{cases} A_{ij}^{k,\ell} = q_i^{k,\ell^T} \text{diag}(c_{ij}^{k,\ell}) k_j^{k,\ell} / \sqrt{d_k} \in \mathbb{R} & \text{if } ij \in E \\ \bar{A}_{ij}^{k,\ell} = \bar{q}_i^{k,\ell^T} \text{diag}(\bar{c}_{ij}^{k,\ell}) \bar{k}_j^{k,\ell} / \sqrt{d_k} \in \mathbb{R} & \text{if } ij \notin E \end{cases} \tag{32}$$

$$Q^{k,\ell} = \begin{bmatrix} h^\ell \\ p^\ell \end{bmatrix} W_Q^{k,\ell}, \ K^{k,\ell} = \begin{bmatrix} h^\ell \\ p^\ell \end{bmatrix} W_K^{k,\ell}, \ V^{k,\ell} = \begin{bmatrix} h^\ell \\ p^\ell \end{bmatrix} W_V^{k,\ell} \in \mathbb{R}^{n \times d_k} \tag{33}$$

$$\bar{Q}^{k,\ell} = \begin{bmatrix} h^\ell \\ p^\ell \end{bmatrix} \bar{W}_Q^{k,\ell}, \ \bar{K}^{k,\ell} = \begin{bmatrix} h^\ell \\ p^\ell \end{bmatrix} \bar{W}_K^{k,\ell}, \ \bar{V}^{k,\ell} = \begin{bmatrix} h^\ell \\ p^\ell \end{bmatrix} \bar{W}_V^{k,\ell} \in \mathbb{R}^{n \times d_k} \tag{34}$$

$$C^{k,0} = e^0 W_e^k, \ \bar{C}^{k,0} = e^0 \bar{W}_e^k \in \mathbb{R}^{n \times n \times d_k} \tag{35}$$

$$\text{and } p_i^{\ell+1} = p_i^\ell + \tanh\left( O_p^\ell \left( \overset{H}{\underset{k=1}{\big\Vert}} \sum_{j \in \mathcal{V}} \frac{w_{p,ij}^{k,\ell}}{\sum_{j' \in \mathcal{V}} w_{p,ij'}^{k,\ell}} v_{p,j}^{k,\ell} \right) \right) \in \mathbb{R}^d, \tag{36}$$

$$w_{p,ij}^{k,\ell} = \begin{cases} \frac{1}{1+\gamma} \cdot \exp(A_{p,ij}^{k,\ell}) & \text{if } ij \in E \\ \frac{\gamma}{1+\gamma} \cdot \exp(\bar{A}_{p,ij}^{k,\ell}) & \text{if } ij \notin E \end{cases}, \tag{37}$$

$$\begin{cases} A_{p,ij}^{k,\ell} = q_{p,i}^{k,\ell \, T} \mathrm{diag}(c_{p,ij}^{k,\ell}) k_{p,j}^{k,\ell} / \sqrt{d_k} \in \mathbb{R} & \text{if } ij \in E \\ \bar{A}_{p,ij}^{k,\ell} = \bar{q}_{p,i}^{k,\ell \, T} \mathrm{diag}(\bar{c}_{p,ij}^{k,\ell}) \bar{k}_{p,j}^{k,\ell} / \sqrt{d_k} \in \mathbb{R} & \text{if } ij \notin E \end{cases} \tag{38}$$

$$Q_p^{k,\ell} = p^\ell W_{p,Q}^{k,\ell}, \ K_p^{k,\ell} = p^\ell W_{p,K}^{k,\ell}, \ V_p^{k,\ell} = p^\ell W_{p,V}^{k,\ell} \in \mathbb{R}^{n \times d_k} \tag{39}$$

$$\bar{Q}_p^{k,\ell} = p^\ell \bar{W}_{p,Q}^{k,\ell}, \ \bar{K}_p^{k,\ell} = p^\ell \bar{W}_{p,K}^{k,\ell}, \ \bar{V}_p^{k,\ell} = p^\ell \bar{W}_{p,V}^{k,\ell} \in \mathbb{R}^{n \times d_k} \tag{40}$$

$$C_p^{k,0} = e^0 W_{p,e}^k, \ \bar{C}_p^{k,0} = e^0 \bar{W}_{p,e}^k \in \mathbb{R}^{n \times n \times d_k} \tag{41}$$

where $W_1^\ell, W_2^\ell \in \mathbb{R}^{d \times d}$, $O^\ell, O_p^\ell \in \mathbb{R}^{d \times d}$, $W_Q^{k,\ell}, W_K^{k,\ell}, W_V^{k,\ell}, \bar{W}_Q^{k,\ell}, \bar{W}_K^{k,\ell}, \bar{W}_V^{k,\ell} \in \mathbb{R}^{2d \times d_k}$, $W_{p,Q}^{k,\ell}, W_{p,K}^{k,\ell}, W_{p,V}^{k,\ell}, \bar{W}_{p,Q}^{k,\ell}, \bar{W}_{p,K}^{k,\ell}, \bar{W}_{p,V}^{k,\ell} \in \mathbb{R}^{d \times d_k}$, $W_e^k, \bar{W}_e^k, W_{p,e}^k, \bar{W}_{p,e}^k \in \mathbb{R}^{d \times d_k}$, and $d_k = d/H$ is the dimension of the $k^{th}$ head for a total of $H$ heads. BN denotes the standard Batch Normalization (Ioffe & Szegedy, 2015). Finally, we make the balance scalar parameter $\gamma \geq 0$ learnable (also clipping its range in $[0, 1]$) differently from (Kreuzer et al., 2021) where its optimal value is computed by grid search.

### C.2.2 GraphiT-LSPE

Similarly to SAN, GraphiT (Mialon et al., 2021) is a full-graph operable Transformer GNN which makes use of the diffusion distance to capture short-range and long-range interactions between nodes depending of the graph topology. This pairwise diffusion distance is used as a multiplicative weight to adapt the weight scores to the closeness or farness of the nodes. For example, if two nodes are close on the graph, them the diffusion distance $K_{ij}$ will have a value close to one, and when the two nodes are far away then the value of $K_{ij}$ will be small.

Unlike SAN, the GraphiT model does not consider separate parameters for existing and non-existing edges for a graph. However, following Kreuzer et al. (2021) and our experiments, separating the parameters for each type of edges showed to improve the performance. Therefore, we augment the original GraphiT architecture with learnable positional features and use two sets of parameters for the edges and the complementary edges to define GraphiT-LSPE. The GraphiT-LSPE model uses the same update equation as SAN-LSPE except for the weight score which are re-defined to introduce the diffusion kernel:

$$w_{ij}^{k,\ell} = \begin{cases} K_{ij} \cdot \exp(A_{ij}^{k,\ell}) & \text{if } ij \in E \\ K_{ij} \cdot \exp(\bar{A}_{ij}^{k,\ell}) & \text{if } ij \notin E \end{cases}. \tag{42}$$

Following (Mialon et al., 2021), the diffusion distance is chosen to be the $p$-step random walk kernel defined as $K = (\mathrm{I}_n - \beta \Delta)^p \in \mathbb{R}^{n \times n}$ where $\mathrm{I}_n, \Delta \in \mathbb{R}^{n \times n}$ is the identity matrix and the graph Laplacian matrix respectively. Hyper-parameter $\beta$ controls the amount of diffusion with value between $[0.25, 0.50]$.

## D  Experiments on non-molecular graphs

We conduct experiments on 3 non-molecular graph datasets to demonstrate the effectiveness of the proposed LSPE architecture on any graph domain in general. We select GatedGCN as the GNN instance here. The datasets used are from the domains of social network (IMDB-BINARY and

Table 4: Results on the IMDB-MULTI, IMDB-BINARY and CIFAR10 superpixels. All scores are averaged over 4 runs with 4 different seeds. On IMDB- each seed experiment is on 10-fold cross validation. **Bold**: GNN's best score. No PosLoss is used with `LSPE`. † denotes the result is taken directly from (Dwivedi et al., 2020).

| Dataset | Model | Init PE | LSPE | $L$ | #Param | TestAcc±s.d. | TrainAcc±s.d. | Epochs | Epoch/Total |
|---|---|---|---|---|---|---|---|---|---|
| IMDB-B | GatedGCN | x | x | 4 | 87122 | 66.050±6.631 | 67.769±4.675 | 115.85 | 0.45s/0.15hr |
| | GatedGCN | RWPE | ✓ | 4 | 91470 | **70.025±5.147** | 72.166±1.706 | 111.17 | 0.56s/0.19hr |
| IMDB-M | GatedGCN | x | x | 4 | 87139 | 45.767±4.906 | 47.725±1.803 | 109.18 | 0.55s/0.17hr |
| | GatedGCN | RWPE | ✓ | 4 | 91483 | **46.467±3.997** | 48.781±1.568 | 100.55 | 0.72s/0.22hr |
| CIFAR10 | GatedGCN | x | x | 4 | 104357 | 67.312±0.311 | 94.553±1.018 | 97.00 | 154.15s/4.22hr † |
| | GatedGCN | RWPE | ✓ | 4 | 107237 | **70.858±0.631** | 78.616±1.006 | 185.00 | 68.51s/3.89hr |

IMDB-MULTI (Morris et al., 2020)) and image superpixels (CIFAR10 (Dwivedi et al., 2020)) with graph classification being the prediction task.

IDBM-BINARY and IMDB-MULTI contain 1,000 and 1,500 graphs respectively which are ego-networks extracted from actor collaboration graphs. There are 2 classes in IMDB-BINARY and 3 classes in IMDB-MULTI with the class denoting the genre of the graph. CIFAR10 is a superpixel dataset of 60,000 graphs where each graph represents a connectivity structure of the image superpixels as nodes. There are 10 classes to be predicted as with the original CIFAR10 image dataset (Krizhevsky et al., 2009).

In Table 4, we show the advantage of using `LSPE` by instantiating GatedGCN−LSPE to train and evaluate on these non-molecular graphs. For evaluation and reporting of results, we follow the respective protocols as specified in Morris et al. (2020); Dwivedi et al. (2020) while comparing two models on a dataset on the similar range of model parameters. In accordance with the molecular datasets (Section 4.2), we consistently observe performance gains on each of the three datasets in Table 4. This result further justifies the usefulness of `LSPE` to be applicable in general for representation learning on any graph domain.

## E  ADDITIONAL MODEL CONFIGURATION DETAILS

In Table 5, additional details on the hyperparameters of different models used in Table 1 are provided. As for hardware information, all models were trained on Intel Xeon CPU E5-2690 v4 server having 4 Nvidia 1080Ti GPUs, with each single GPU running 1 experiment which equals to 4 parallel experiments on the machine at a single time.

Table 5: Additional hyperparamters for the models used in Table 1. $k$ is the dimension of PE, or the steps of random walk if the PE is RWPE. $\beta$ and $p$ is applicable to GraphiT (Sec. C.2.2). **Init_lr** and **Min_lr** are the initial and final learning rates for the learning rate decay strategy where the lr decays with a reduce **Factor** if the validation score doesn't improve after the **Patience** number of epochs. $\alpha$ and $\lambda$ are applicable when PosLoss is used (Eqn. 12).

| | Model | Init PE | `LSPE` | PosLoss | $k$ | $\beta$ | $p$ | Init_lr | Patience | Factor | Min_lr | $\alpha$ | $\lambda$ |
|---|---|---|---|---|---|---|---|---|---|---|---|---|---|
| **ZINC** | GatedGCN | **x** | **x** | **x** | - | - | - | 1e-3 | 25 | 0.5 | 1e-6 | - | - |
| | GatedGCN | **LapPE** | **x** | **x** | 8 | - | - | 1e-3 | 25 | 0.5 | 1e-6 | - | - |
| | GatedGCN | **RWPE** | ✔ | **x** | 20 | - | - | 1e-3 | 25 | 0.5 | 1e-6 | - | - |
| | GatedGCN | **RWPE** | ✔ | ✔ | 20 | - | - | 1e-3 | 25 | 0.5 | 1e-6 | 1 | 1e-1 |
| | PNA | **x** | **x** | **x** | - | - | - | 1e-3 | 25 | 0.5 | 1e-6 | - | - |
| | PNA | **RWPE** | ✔ | **x** | 16 | - | - | 1e-3 | 25 | 0.5 | 1e-6 | - | - |
| | SAN | **x** | **x** | **x** | - | - | - | 3e-4 | 25 | 0.5 | 1e-6 | - | - |
| | SAN | **RWPE** | ✔ | **x** | 16 | - | - | 7e-4 | 25 | 0.5 | 1e-6 | - | - |
| | GraphiT | **x** | **x** | **x** | - | 0.25 | 16 | 3e-4 | 25 | 0.5 | 1e-6 | - | - |
| | GraphiT | **RWPE** | ✔ | **x** | 16 | 0.25 | 16 | 7e-4 | 25 | 0.5 | 1e-6 | - | - |
| | **Model** | **Init PE** | `LSPE` | **PosLoss** | $k$ | $\beta$ | $p$ | **Init_lr** | **Patience** | **Factor** | **Min_lr** | $\alpha$ | $\lambda$ |
| **MOLTOX21** | GatedGCN | **x** | **x** | **x** | - | - | - | 1e-3 | 25 | 0.5 | 1e-5 | - | - |
| | GatedGCN | **LapPE** | **x** | **x** | 3 | - | - | 1e-3 | 25 | 0.5 | 1e-5 | - | - |
| | GatedGCN | **RWPE** | ✔ | **x** | 16 | - | - | 1e-3 | 25 | 0.5 | 1e-5 | - | - |
| | PNA | **x** | **x** | **x** | - | - | - | 5e-4 | 10 | 0.8 | 2e-5 | - | - |
| | PNA | **RWPE** | ✔ | **x** | 16 | - | - | 5e-4 | 10 | 0.8 | 2e-5 | - | - |
| | PNA | **RWPE** | ✔ | ✔ | 16 | - | - | 5e-4 | 10 | 0.8 | 2e-5 | 1e-1 | 100 |
| | SAN | **x** | **x** | **x** | - | - | - | 7e-4 | 25 | 0.5 | 1e-6 | - | - |
| | SAN | **RWPE** | ✔ | **x** | 12 | - | - | 7e-4 | 25 | 0.5 | 1e-6 | - | - |
| | GraphiT | **x** | **x** | **x** | - | 0.25 | 16 | 7e-4 | 25 | 0.5 | 1e-6 | - | - |
| | GraphiT | **RWPE** | ✔ | **x** | 16 | 0.25 | 16 | 7e-4 | 25 | 0.5 | 1e-6 | - | - |
| | **Model** | **Init PE** | `LSPE` | **PosLoss** | $k$ | $\beta$ | $p$ | **Init_lr** | **Patience** | **Factor** | **Min_lr** | $\alpha$ | $\lambda$ |
| **MOLPCBA** | GatedGCN | **x** | **x** | **x** | - | - | - | 1e-3 | 25 | 0.5 | 1e-4 | - | - |
| | GatedGCN | **LapPE** | **x** | **x** | 3 | - | - | 1e-3 | 25 | 0.5 | 1e-4 | - | - |
| | GatedGCN | **RWPE** | ✔ | **x** | 16 | - | - | 1e-3 | 25 | 0.5 | 1e-4 | - | - |
| | PNA | **x** | **x** | **x** | - | - | - | 5e-4 | 4 | 0.8 | 2e-5 | - | - |
| | PNA | **RWPE** | ✔ | **x** | 16 | - | - | 5e-4 | 10 | 0.8 | 2e-5 | - | - |

