# OpenReview forum: "Graph Neural Networks with Learnable Structural and Positional Representations"
_ICLR.cc/2022/Conference — ICLR 2022 Poster_

### Official Review · Reviewer_ENFN · 2021-10-29

**Correctness:** 3
**Technical Novelty And Significance:** 3
**Empirical Novelty And Significance:** 3
**Recommendation:** 6
**Confidence:** 3

**Main Review:**

(Sorry, I re-organized the review to meet the guideline)

### points
(+) motivation sounds reasonable

(+) easy read, appendix full of useful information and explanations

(+) technical contents including formulations are easy to follow

(-) Significance of experimental improvements: are they really significant?

(-) Initialize of PE and the LSPE architecture, which item is important?

### comments

The manuscript is easy-to-read. I feel no difficulty in understanding the discussions in the manuscript.
Also, the appendix provides many useful material to readers, an additional plus (excepting the Section C.2. it is quite complicated lines of equations).
I am rather new to the PEs for GNNs, but the manuscript helps me understanding the current PE research status and how we can place this work in that context.

I like the idea of allowing PE to be updated through layers. As the manuscript claims, the PE is an important cue to distinguish isomorphism in graphs, but the initial injection of PEs like in Eqs(4-6) is not powerful enough to obtain good embedding features.
The proposed formulation Eqs.(7-9) are clear to understand.

The discussions in Appendix A is powerful enough for me to feel "in practice" the RWPE will work fine, though the proposed RWPE(+LSPE) has no theoretical guarantee to distinguish isomorph nodes.

My main concern is related to the experimental results.

Tables 1,2:
I agree that the LSPE works fine to reduce the regression MAEs on the ZINC dataset.
However, the improvements in the remaining MOLTOX21 and the MOPCBA datasets seems marginal.
Considering the standard deviations (thank you authors to provide the s.d. !!), I'm not sure whether these improvements are statistically significant.
Given the current manuscript, I need to judge that the proposed LSPE is successful in show solid improvements against the existing PE works in the minority (1 out of 3) of datasets.

Some ideas:
One is to conduct statistical tests on the current scores. If the LSPE obtains small p-values in the MOL datasets, then, it is great!
Another is to test more diverse datasets. If the LSPE is really powerful (I believe so), the LSPE will achieve clear score improvements in many additional datasets.

Perhaps adding extra metrics to support the efficacy of the LSPE may improve the impression of the manuscript.
For example, report memory/space usages of the proposed model.
One important motivation for this research is a computational difficulty of stronger GNNs (than 1-WL).
Then it may be beneficial to show that the existing approach actually does not work with larger graphs.

Table 3:
The results in the Table 3 indicates that the choice of the initial PE (LapPE, RWPE) is much important than the LSPE updates.
Perhaps it may be straightforward to emphasize the importance of the PE Initialization, not the LSPE architecture?
In my understanding, the current manuscript weighs more on the LSPE (e.g. the subsection "contribution" in Sec. 1 and Fig 1).


**Summary Of The Paper:**

This paper is concerning the Positional Encoding (PE) for GNNs.
PE augments the typical GNNs to distinguish isomorphic nodes. However, existing PE models such as Laplacian eigenvectors require huge computational resources.
This manuscript proposes LSPE that augments the input to the nodes AND the embedding vectors with PE elements.
The LSPE models iteratively updates the embedding for PEs, in addition to the node feature embeddings.
The update formula of the PE embedding is similar to those of the node feature embedding, thus the computational requirements of the LSPE does not .
The manuscript tests two ways of PEs, one is based on the Laplacian, and the other is based on the random-walk. The manuscript also proposes the PE=only loss to foster the training.
Experimental results shows that the propose LSPE can improve the graph regression of the ZINC dataset greatly, and also can achieve some improvements in graph classification tasks on the MOLTOX21 and the MOLPCBA datasets.


**Summary Of The Review:**

I find the proposed idea reasonable and straightforward (in a positive manner). But I'm not fully sure the experimental results well support the claims. I also have a concern that the most influential time is the PE init., not the PE embed. update formulation.


============================================

After author feedbacks:

I feel the answers from the authors are largely satisfactory.

Other reviewers raise a concern about the novelty of the proposed methods. I agree that the novelty is minor.

However, the experimental results and the statistical test reports indicate the strongness of the proposed framework in practice.
Additional experiments on other domains are a positive surprise.
Therefore I modify the review score one step upward.

---

> ### Author Response · Authors · 2021-11-21
> **Response to Reviewer ENFN (Part 1/2)**
>
> We thank the reviewer for her/his careful reading and comments regarding our work. Please, see below our answer to the raised comments/questions.
>
> >**Reviewer**: My main concern is related to the experimental results. Tables 1,2: I agree that the LSPE works fine to reduce the regression MAEs on the ZINC dataset. However, the improvements in the remaining MOLTOX21 and the MOPCBA datasets seems marginal. Considering the standard deviations (thank you authors to provide the s.d. !!), I'm not sure whether these improvements are statistically significant. Given the current manuscript, I need to judge that the proposed LSPE is successful in show solid improvements against the existing PE works in the minority (1 out of 3) of datasets. [...] Some ideas: One is to conduct statistical tests on the current scores. If the LSPE obtains small p-values in the MOL datasets, then, it is great! Another is to test more diverse datasets. If the LSPE is really powerful (I believe so), the LSPE will achieve clear score improvements in many additional datasets.
>
> **Authors**: As suggested, we conducted a statistical test using the standard p-value test. The p-values for ZINC, MOLTOX21 and MOPCBA are 0.00011, 0.0010 and 0.0078 respectively. As a test is considered to be statistically significant for p<0.05, so are the reported results.
>
> We also ran experiments on additional non-molecular datasets from social network domain (IMDB-BINARY, IMDB-MULTI) and image superpixel graphs (CIFAR10). We have added the results and discussion in the supplementary Section D. We observed the proposed LSPE enhances the GatedGCN consistently on all the 3 non-molecular datasets, from a 66.05% class accuracy score to 70.02% for IMDB-BINARY, from 45.76% to 46.46% for IMDB-MULTI and from 67.31% to 70.85% for CIFAR10. Note here that we did not perform exhaustive hyperparameter search on these experiments but rather compare fairly the two models (with and without LSPE) on the same parameter budget to show the usefulness of our simple and yet effective approach for GNNs.
> ___
> >**Reviewer**: Perhaps adding extra metrics to support the efficacy of the LSPE may improve the impression of the manuscript. For example, report memory/space usages of the proposed model. One important motivation for this research is a computational difficulty of stronger GNNs (than 1-WL). Then it may be beneficial to show that the existing approach actually does not work with larger graphs.
>
> **Authors**: The memory/space usage is O(n) for sparse MP-GNNs-LSPE and O(n^2) for fully-connected Transformer-GNNs. As observed by the reviewer, this is an important computational advantage for the proposed LSPE framework compared to higher-order GNNs (that have higher k-WL power, k>=3). While we have not reported the memory/space usage in our paper, we did report the times per epoch and in total, which we believe illustrate well the computational efficiency of the method.
>
> As an instance of memory usage, we consider ZINC with a budget of 500k for fair comparison between the proposed GatedGCN-LSPE model (> 1-WL power) and the 3WL-GNN architecture developed in [Maron et al, Provably powerful graph networks, 2019] with a proven power of 3-WL. The memory usage for the learnable parameters of GatedGCN-LSPE and 3WL-GNN models are approximatively the same with 2.1MB and 2.0MB respectively, which is consistent with a budget of 500k parameters for both models. The memory usage for a mini-batch of 512 graphs is 23MB for a single GatedGCN-LSPE layer and 1046MB for a 3WL-GNN layer.
> ___
> >**Reviewer**: The results in the Table 3 indicates that the choice of the initial PE (LapPE, RWPE) is much important than the LSPE updates. Perhaps it may be straightforward to emphasize the importance of the PE Initialization, not the LSPE architecture? In my understanding, the current manuscript weighs more on the LSPE (e.g. the subsection "contribution" in Sec. 1 and Fig 1).
>
> **Authors**: We agree the choice of the PE initialization is an important aspect of the framework. However, learnable positional representation at every layer can also offer a clear advantage for example with ZINC, improving from 0.122 to 0.100 when updating the positional information. Our motivation is to propose a general framework to decouple and learn as much as possible structural and positional information in GNNs. The optimal choice of the initial PE, the structural and positional aggregation functions will depend on the data and the task at hand, but we do not want to limit our framework and we let it as flexible as possible including the learnable positional representation.
> ___

---

> > ### Author Response · Authors · 2021-11-21
> > **Response to Reviewer ENFN (Part 2/2)**
> >
> > >**Reviewer**: The discussions in Appendix A is powerful enough for me to feel "in practice" the RWPE will work fine, though the proposed RWPE(+LSPE) has no theoretical guarantee to distinguish isomorph nodes.
> >
> > **Authors**: The proposed RWPE+LSPE model is theoretically strictly more powerful than the 1-WL test as it is able to distinguish non-isomorphic graphs like CSL graphs, and Decalin and Bicyclopentyl graphs (see Sections A.1 and A.2). The model is also guaranteed to distinguish non-isomorphic nodes which have a unique k-hop topological neighborhood for a sufficient large value k. We discussed in details this condition in Section A.3. While this condition is experimentally true for most nodes, it is not always satisfied. But despite this approximation, for a sufficiently large number k of random walk iterations, RWPE is still able to capture global higher-order positioning of nodes that is used as initial PE, and is beneficial to the proposed LSPE architecture as demonstrated by the gain of performance in several experiments.
> > ___
> > >**Reviewer**: I find the proposed idea reasonable and straightforward (in a positive manner). But I'm not fully sure the experimental results well support the claims. I also have a concern that the most influential time is the PE init., not the PE embed. update formulation.
> >
> > **Authors**: We propose a generic and effective framework to improve the representation power and test performance of any message-passing GNN. Per the reviewer’s suggestion, we tested the statistical significance of the proposed technique and experimented on three new non-molecular datasets. We also discussed our motivation to keep a general framework to decouple and learn positional and structural information for future work. We hope to have addressed most of your concerns. Let us know if you have any other questions.

---

> > > ### Comment · Reviewer_ENFN · 2021-11-22
> > > **Thank you for the clarifications**
> > >
> > > Thank you for the detailed clarification (to all reviewers' comments) !!
> > > I'm happy to hear that the statistical test proves the efficacy of the proposed model.
> > > Additional experiments on other domains are positive surprise.
> > >
> > > In this short time I cannot follow all your answers. I will modify my evaluations after reading the revised paper and your answers to reviewers' questions.
> > >
> > > Thank you for your hard works!

---

> > > > ### Author Response · Authors · 2021-11-23
> > > > **Thank you**
> > > >
> > > > Thank you. Your positive feedback is very much appreciated.

---

> ### Author Response · Authors · 2021-11-30
> **Response to Reviewer ENFN post feedback on rebuttal**
>
> >**Reviewer** : Other reviewers raise a concern about the novelty of the proposed methods. I agree that the novelty is minor.
>
> **Authors**: We would like to thank very much the reviewer for her/his feedback on our rebuttal.
>
> The novelty of the proposed method seems to be of concern. We would like to clarify that the idea of decoupling structural and positional representations has never been published in the literature to the best of our knowledge.
>
> At first sight, it would seem a minor novelty but we would like to emphasize that this idea is based on a deep understanding of a major limitation of most GNN methods (that is the low representation power due to the inability of structural GNNs to differentiate simple graph symmetries). By decoupling structure and position information in GNN, the network is able to improve its performance by capturing separately these two properties.
>
> This key observation can be used for any GNN and the positional information is also general and is not limited to LapPE or RWPE that we used in this paper. In other words, we present a novel general framework that can augment any existing GNN performance, and consequently may have an impact into the GNN literature as a simple but yet effective idea.
>
> We also consider this new idea as a step forward into the benefit of positional encoding introduced in the standard Transformers of Vaswani et al. 2017, where the PE was critical to inform the network about the positions of the words. For this work, the PE is critical for the node position, and must be learned due to the complex graph topology.
>
> We will make the presentation of the novelty of the proposed method more explicit in the revised manuscript.

---

### Official Review · Reviewer_2bUV · 2021-10-31

**Correctness:** 4
**Technical Novelty And Significance:** 3
**Empirical Novelty And Significance:** 3
**Recommendation:** 8
**Confidence:** 4

**Main Review:**

Pros:
- The idea of disentangling the positional representation is novel.
- The empirical results are really impressive.
- The paper is well-organized and the writing is good.

Concerns:
- The design of this method is complicated and may introduce more parameters. This may cause problems that the model may tend to overfit the data and make it unclear if the improvement comes from the increase of the number of parameters (I wonder why the #Param does not increase so much as listed in the tables as the sizes of A_i's in (14) all double?).
- How the positional encoding proposed in the paper compares to relative positional encoding in GNN, e.g., that used in Graphormer (Ying et al. 2021)? And is it possible to further boost the performance by combining it?

**Summary Of The Paper:**

This paper proposes to use a separate positional representation with its own loss function for graph representation learning and achieve impressive empirical results.

**Summary Of The Review:**

The proposed method has strong motivation and empirical results and I vote for an acceptance.

---

> ### Author Response · Authors · 2021-11-21
> **Response to Reviewer 2bUV**
>
> We thank the reviewer for her/his careful reading and comments regarding our work. Please, see below our answer to the raised comments/questions.
>
> >**Reviewer**: The design of this method is complicated and may introduce more parameters. This may cause problems that the model may tend to overfit the data and make it unclear if the improvement comes from the increase of the number of parameters (I wonder why the #Param does not increase so much as listed in the tables as the sizes of A_i's in (14) all double?).
>
> **Authors**: We would like to clarify the design of the method, particularly Eqns (7-9). The main goal of LSPE is to decouple structural representation (standard Message Passing-GNNs) and positional representation (nodes in arbitrary graphs have no ordering and this diminishes the representation power of GNNs). To achieve this, we simply introduce two vectors h and p representing the structure and position respectively. In addition, vector e represents the edge information. Then, we design aggregation functions $f_h, f_p, f_e$ to update the ($h, p, e$) representation at the next layer. And that is all for the general method. Later, we instantiated this general framework to a specific GNN like GatedGCN or PNA where we used their specific $f_h, f_p$ aggregation functions. In summary, the design is quite generic, and the proposed framework has been effective to improve the performance of respective GNNs.
>
> To demonstrate that the gain is not attributed to adding more parameters to the network, we followed the same benchmarking protocol introduced in [Dwivedi et al., 2020] to fairly compare all models on ZINC. For ZINC, all the scores in Table 2a and Table 1 are produced with a fixed number of 500k model parameters. This allows to impartially compare all models, particularly the two models without and with LSPE; such as GatedGCN-NoPE (with a MAE of 0.251) and GatedGCN-RWPE-LSPE (with 0.093 MAE), and shows that the gain is not coming from additional parameters but rather from the new ideas of decoupling structural and positional information, using a powerful PE (with RWPE) and updating simultaneously positional and structural representation in the GNNs.
>
> To make sure the two models, GatedGCN-NoPE and GatedGCN-RWPE-LSPE, have the same number of parameters, we decreased the hidden feature dimension when LSPE is used (for example on ZINC, please refer to the difference in the ‘hidden_dim’ of GatedGCN-NoPE and GatedGCN-RWPE-LSPE in the ‘configs/’ directory of our source code commented anonymously in this forum. These are respectively 154 and 118 equalling model parameters in ~500k for both models enabling fair comparison).
>
> This clarifies that the performance gain comes from our new architectural contribution, and not from the increase in model parameters (as we do not double the total number of parameters).
> ___
> >**Reviewer**: How the positional encoding proposed in the paper compares to relative positional encoding in GNN, e.g., that used in Graphormer (Ying et al. 2021)? And is it possible to further boost the performance by combining it?
>
> **Authors**: The positional encoding introduced in Graphormer (Ying et al. 2021) and also in GraphiT (Mialon et al. 2021) is of different nature as it is a relative distance encoding representation (it represents a distance between two nodes in a graph) and our positional encoding is absolute (such as the coordinates of a pixel in an image). Our numerical experiments showed that absolute positional encoding performs better than relative ones. For example for ZINC, our proposed architecture (0.093 MAE score) performs significantly better than Graphormer (0.122) and GraphiT (0.181). Besides, we investigated the idea to further boost the performance of our framework by combining the relative positional encoding of GraphiT and the absolute position with the GraphiT-LSPE model, see particularly Eqn (42) with $K_{ij}$, the diffusion distance between two nodes. The performance of GraphiT-LSPE was 0.106, better than the original GraphiT, but not as good as solely using the absolute positional encoding which provides a 0.093 score with GatedGCN-LSPE.
> ___
> **Authors**: We hope to have clarified most of your concerns. Let us know if you have any other questions.

---

### Official Review · Reviewer_vVMd · 2021-11-01

**Correctness:** 3
**Technical Novelty And Significance:** 3
**Empirical Novelty And Significance:** 3
**Recommendation:** 8
**Confidence:** 4

**Main Review:**

**Strengths**

* The problem of message passing neural networks(MPNNs) lacking positional information of nodes is well-motivated by molecules.
* The Proposal is understandable & straightforward, with an affordable memory cost.
* Well written and provides detailed related works.

The idea of decoupling structural and positional encodings in MPNNs while keeping in linear complexity is very simple yet a clear idea to me.  Also, the motivation of this paper is elucidated via molecules where learned node embeddings do not differentiate between positions. Finally, the related works section is well written and explains the problems in the GNN domain clearly.

**Weaknesses**
* I would expect more experiments on different domains (such as recommendation systems), not just for the molecular domain.
Introduced PosLoss does not introduce an important change to results at all. Thus, the existence of this newly introduced loss seems questionable.
* Claim that achieving SOTA results for the ZINC dataset(12K graphs) seems questionable because [1] achieves a 0.074 MAE score.
* For the large-scale graph dataset( around 430K graphs), OGBG-MOLPCBA, which is bigger in the number of graphs and more complicated in terms of learning, the benefits are questionable. It seems the choice of aggregation outweighs the effect of positional encoding[2,3].


My main concern is that the paper claims it is doing more than the listed contributions. First of all, introducing loss for positional encodings seems not working at all. Second, the claim is that achieving SOTA results for the ZINC dataset(12K graphs) seemed false to me as there is a recent paper [1] achieving 0.074 mean-absolute error with edge features. Also, doing more experiments on different domains, I think, is needed to show that this works. Idea working on the molecular domain only does not imply that this idea works well. Experimenting on domains like knowledge graphs would be nice to see its results since the position of nodes also plays a significant role in that domain.

Finally, the most important thing for me is the benefit of learning decoupled positional embeddings in graph neural networks. Graph neural network architectures such as Directional Graph Networks[2] and PHC-GNN[3] seem to outperform the proposed model on the OGBG-MOLPCBA dataset. Even leveraging LSPE on Principal Neighborhood Aggregation[4] gives incremental results. What if we add more aggregators to PNA or use different domains like in PHC-GNN? I would expect a discussion on why these methods outperform the proposed plan as I believe it is important to tell readers why these methods outperform yours. Overall, it is a good paper with a simple idea, but I would expect more reasoning on why this method can fail and more experiments on different domains.

[1]Bodnar, Cristian, et al. "Weisfeiler and Lehman go cellular: Cw networks." arXiv preprint arXiv:2106.12575 (2021).

[2] Beaini, Dominique et al. “Directional Graph Networks.” ICML (2021).

[3] Le, Tuan, et al. "Parameterized hypercomplex graph neural networks for graph classification." arXiv preprint arXiv:2103.16584 (2021).

[4] Corso, Gabriele, et al. "Principal neighborhood aggregation for graph nets." arXiv preprint arXiv:2004.05718 (2020).

**Summary Of The Paper:**

This paper introduces the idea of learning positional representations to enrich node representations of graph neural networks(GNN) to improve the representation power of message-passing graph neural networks(MPNNs) while keeping in linear complexity.

**Summary Of The Review:**

Overall, this paper tries to alleviate a significant problem in MPNNs. Still, limited experiments in a single domain and counter-intuitive results for OGBG-MOLPCBA and ZINC datasets make this paper’s contributions questionable.

---

> ### Author Response · Authors · 2021-11-21
> **Response to Reviewer vVMd (Part 1/2)**
>
> We thank the reviewer for her/his careful reading and comments regarding our work. Please, see below our detailed answer to the raised comments/questions.
>
> >**Reviewer**: I would expect more experiments on different domains (such as recommendation systems), not just for the molecular domain.
>
> **Authors**: As suggested, we conducted additional experiments using GatedGCN-LSPE on 3 non-molecular datasets from social network domain (IMDB-BINARY, IMDB-MULTI) and image superpixel graphs (CIFAR10). We have added the results and discussion in the supplementary Section D. We observed the proposed LSPE enhances the GatedGCN consistently on all the 3 non-molecular datasets, from a 66.05% class accuracy score to 70.02% for IMDB-BINARY, from 45.76% to 46.46% for IMDB-MULTI and from 67.31% to 70.85% for CIFAR10. Note here that we did not perform exhaustive hyperparameter search on these experiments but rather compare fairly the two models (with and without LSPE) on the same parameter budget to show the usefulness of our simple and yet effective approach for GNNs.
> ___
> >**Reviewer**: Introduced PosLoss does not introduce an important change to results at all. Thus, the existence of this newly introduced loss seems questionable.
>
> **Authors**: We agree that the introduction of the positional loss only shows a slight improvement in the case of ZINC (from 0.093 to 0.090) and no improvement for the other datasets. Our motivation to use an additional positional loss comes from the domain of Manifold Learning, where the goal is to embed the graph into a lower-dimensional space which can be meaningful and helps improving performance. Unfortunately, the proposed Dirichlet loss coupled with the orthogonality term did not improve the performance. However, we thought that suggesting the use of a positional loss in our paper might be of interest to the readers who could later define a more appropriate loss by leveraging ideas from Manifold Learning.
> ___
> >**Reviewer**: Claim that achieving SOTA results for the ZINC dataset (12K graphs) seems questionable because [1] achieves a 0.074 MAE score.
>
> **Authors**: We acknowledged [1] in the submission and the discussion section of our experiments *“We note here that ZINC scores can even be boosted beyond LSPE’s SOTA when expert prior knowledge is used (Bouritsas et al., 2020; Bodnar et al., 2021)”*. We did not directly compare to [1] because [1] is a domain-expert GNN while our GNN is designed to be domain-agnostic, that is without making any specific assumption on the class of graphs at hand. For example in ZINC, the predictive score is directly correlated to the number of rings in the molecule. The model [1] (CIN) uses this expert prior information to define their GNN architecture, and achieved 0.074 MAE score.
>
> An ablation study in Appendix E.5 of [1] shows that the performance of CIN degrades to 0.159 without the use of the ring structure. Besides, the 0.074 MAE score reporting in the paper is not obtained with the budget of 500k parameters (defined in Dwivedi et al. (2020) for fair comparison on ZINC) which is used in Table 2(a) for all GNN models, but with 1.7M parameters (3x more) when running their official code. The increase in the number of parameters may help to improve the score.
>
> We also ran the CIN model on the PCBA dataset with the same official code and 6.8M model parameters comparable with PNA-LSPE (we have not performed hyper-parameter search). The average precision of CIN for this dataset is 0.273 for 32.51hr training time, which does not overcome the PNA-LSPE, DGN and PHC-GNN models with 0.287, 0.288 and 0.294 respectively. Finally, leveraging graph structures like rings increases the complexity of the GNN design and code implementation, unlike our framework which can augment any existing GNN while keeping a linear complexity.
>
> Following the reviewer’s comment, we have changed in the paper the line “LSPE improves the state-of-the-art” into “LSPE improves the state-of-the-art for domain-agnostic GNNs” to make it clear.
> ___

---

> > ### Author Response · Authors · 2021-11-21
> > **Response to Reviewer vVMd (Part 2/2)**
> >
> > >**Reviewer**: For the large-scale graph dataset (around 430K graphs), OGBG-MOLPCBA, which is bigger in the number of graphs and more complicated in terms of learning, the benefits are questionable. It seems the choice of aggregation outweighs the effect of positional encoding [2,3]. [...] Overall, it is a good paper with a simple idea, but I would expect more reasoning on why this method can fail.
> >
> > **Authors**: Indeed, the PHC-GNN [3] and DGN [2] models achieve a better performance than the reported score of proposed PNA-LSPE for OGBG-MOLPCBA. Decoupling structural and positional information is a general idea that leverages the theoretical work of [Murphy et al., 2019], which proved a higher presentation power for GNNs when considering positional encoding. However, getting better representation power does not necessarily translate into better generalization performance as shown in [Dwivedi et al. (2020)] where GIN (1-WL provable) does not surpass GAT or GatedGCN. The same result applies here. For some datasets and tasks, the proposed LSPE framework may not improve the original model either because the original aggregation functions are good enough or the node feature contain enough discriminative information to alleviate the need of additional PE. This being said, for some models like PNA, we can see an increase of performance by 2.87%, which is a significant gain compared to the other techniques because the task is hard (the p-score is 0.0078 for PNA-LSPE).
> >
> > Additionally, for PHC-GNN, we noted that there were only 2 domains where the model was evaluated; molecular datasets and superpixel graphs (MNIST, CIFAR10). For the superpixel graphs, PHC-GNN does not surpass the SOTA and lags the best score on CIFAR10. We refer to the leaderboard at [5]. From our additional experiments on 3 non-molecular datasets in supplementary section D, GatedGCN-LSPE gives 70.85% test accuracy on CIFAR10 compared to PHC-GNN’s 66.80%. At the same time, the score of PHC-GNN on ZINC is 0.164 (PHC-5 in [3]) and takes 33 seconds per epoch whereas GatedGCN-LSPE gives 0.093 at 15 seconds per epoch. Comparing PHC-GNN with LSPE on ZINC is thus computationally efficient and empirically more accurate.
> > ___
> > >**Reviewer**: Overall, this paper tries to alleviate a significant problem in MPNNs. Still, limited experiments in a single domain and counter-intuitive results for OGBG-MOLPCBA and ZINC datasets make this paper’s contributions questionable.
> >
> > **Authors**: We propose a simple and yet effective framework to improve the representation power and test performance of any message-passing GNN. Per the reviewer’s suggestion, we conducted additional experiments on three non-molecular datasets to show the generality of the proposed technique. We also discussed the domain-agnostic vs domain-expert property of the proposed framework compared to CIN. And we talked over the limitations of the proposed technique by comparing to PHC-GNN and DGN. We hope our answers have addressed your concerns. Let us know if you have any other questions or concerns.
> > ___
> > *References*:
> > [1] Bodnar, Cristian, et al. "Weisfeiler and Lehman go cellular: Cw networks." arXiv preprint arXiv:2106.12575 (2021).
> > [2] Beaini, Dominique et al. “Directional Graph Networks.” ICML (2021).
> > [3] Le, Tuan, et al. "Parameterized hypercomplex graph neural networks for graph classification." arXiv preprint arXiv:2103.16584 (2021).
> > [4] Corso, Gabriele, et al. "Principal neighborhood aggregation for graph nets." arXiv preprint arXiv:2004.05718 (2020).
> > [5] CIFAR10 Leaderboard: https://github.com/graphdeeplearning/benchmarking-gnns/blob/master/docs/07_leaderboards.md#5-cifar10---graph-classification

---

### Official Review · Reviewer_E8yv · 2021-11-02

**Correctness:** 3
**Technical Novelty And Significance:** 2
**Empirical Novelty And Significance:** 2
**Recommendation:** 5
**Confidence:** 4

**Main Review:**

Strengths:

1. consider postional embedding in GNNs, the proposed framework is very general and can be directly used in many existing GNNs.

2. The design is simple, insert the positional embedding to current existing message passing function. And also update the positional embedding as well e.g., in Eq.(9). Some method like Lapalacian or Random Walk are used to initilize the positional embedding.

3. The results are good in the experiements.

Weaknesses:

1. My big concern is that the motivation of positional embedding is not very strong. For example, Eq.(7) can do the similar thing as Eq.(9), just using the vector concatenation. As claimed in the paper, the main different is that the Eq.(9) use the tanh function. Does that mean the Tanh is key for the positional embedding? Experiments on different activation function may be more convicing.

2. The author claim that "PE as eigenvectors are defined up to ±1, leading to 2k number of possible sign values". However,  the use of the tanh activation function to allow positive and negative values for the positional coordinates. Will the positive and negative sign also casue the same issue?

3. Some other baselines can be comapred, like
Jiaxuan You, Rex Ying, and Jure Leskovec. Position-aware graph neural networks. In International
Conference on Machine Learning, pp. 7134–7143. PMLR, 2019.


**Summary Of The Paper:**

In this work, the authors mainly focus on the node positinal embedding for GNN. The proposed positional embedding is geneal and can be applied to many GNNs.

**Summary Of The Review:**

In summary, I think the proposed method is this work is very practical. It seems that using a simple positional embedding help a lot. However, it seems that the motivation is not very strong, like how to explain this positional embedding.

---

> ### Author Response · Authors · 2021-11-21
> **Response to Reviewer E8yv (Part 1/2)**
>
> We thank the reviewer for her/his careful reading and comments regarding our work. We would like to take this opportunity to clarify the use of the positional encoding.
>
> >**Reviewer**: My big concern is that the motivation of positional embedding is not very strong.
>
> **Authors**: Positional encoding (PE) have been proven essential to increase the performance of GNNs. Standard MP-GNNs have a low representation power, and fail to distinguish simple graph structures. An approach to improve the GNN representation is to design k-order GNNs, which require the use of k-tuple of nodes and hence increase the complexity to O(n^k) while not necessarily improving the generalization performance. An alternative is to add positional encoding (PE) to the graph vertices and it was shown in [Murphy et al., 2019] that the PE theoretically increase the representation power of any GNN while keeping a linear complexity. This result serves as a theoretical foundation and justification of using PE in GNNs. But any PE would not work well s.a. the index PE of [Murphy et al., 2019], which has led us to propose two novel contributions in this line of work; (1) the use the diagonal of the k-step Random Walk Matrix as a discriminative and low-complexity PE (which leads to GNNs strictly more powerful than the 1-WL test as shown in Sections A.1 and A.2), (2) the update of PE at each layer to better adapt to the task at hand. This results in a very simple and powerful framework that can learn both structural and positional information and can be applied to any GNN.
>
> For an intuitive understanding, consider a molecule where the location of a particular graph structure such as a ring (cycle of atoms) is key at predicting a label in a given task. Standard GNNs that only update structural representation will fail to solve this task [Chen et al., 2020], whereas our proposed architecture would succeed thanks to the positional information. Next, consider the task of detecting a ring of carbon atoms inside a molecule containing both a ring of carbon atoms and a ring of hydrogen atoms. In this case as well, our proposed architecture is well-equipped to solve this task as the network decouples the structural information (the type of atoms; carbon or hydrogen) and the positional information (that is necessary to detect cycles) and combine them by simple concatenation at the final layer. We believe the interest of proposing such an architecture is demonstrated in the experiments and further justified in Section 4.3.
> ___
> >**Reviewer**: For example, Eq.(7) can do the similar thing as Eq.(9), just using the vector concatenation.
>
> **Authors**: Indeed, Eqns. (7) and (9) follow the same analytical form of the message-passing equivariant function. But the key idea here is to decouple/disentangle the structural and positional representations to make them easy to learn by the network, unlike existing GNNs that use PE but mix structure and position into the first layer. By decoupling structure representation in Eqn (7) (initialized with node attributes) and position representation in Eqn (9) (initialized with RWPE), the network is able to optimize their representation for the task at hand. We use the same message-passing analytical form for Eqns (7) and (9) for simplicity (and this approach is demonstrated to be very effective in the experiments section), but different functions can be used if additional domain knowledge is available and more appropriate aggregation functions are known.
> ___
> >**Reviewer**: As claimed in the paper, the main different is that the Eq.(9) use the tanh function. Does that mean the Tanh is key for the positional embedding? Experiments on different activation function may be more convicing.
>
> **Authors**: The presence of tanh function as non-linear activation function is to allow both positive and negative values for the positional representation. That can be helpful when learning positional representation like Laplacian PE where the eigenvectors have positive and negative values. Nevertheless, we ran, as suggested, an experiment for ZINC with ReLU activation in p-update: the MAE is 0.093 with tanh versus 0.099 with ReLU. All in one, the choice of the non-linearity for the positional representation is not as critical as the decoupling of the structural and positional information, and the choice of the initial PE.
> ___
> >**Reviewer**: The author claim that "PE as eigenvectors are defined up to ±1, leading to 2k number of possible sign values". However, the use of the tanh activation function to allow positive and negative values for the positional coordinates. Will the positive and negative sign also casue the same issue?
>
> **Authors**: We would like to clarify that the tanh activation function and the sign ambiguity of Laplacian eigenvectors are unrelated. As written above, the use of tanh function is to allow positive and negative values for the positional coordinates. Therefore, it does not cause any sign ambiguity.
> ___

---

> > ### Author Response · Authors · 2021-11-21
> > **Response to Reviewer E8yv (Part 2/2)**
> >
> > >**Reviewer**: Some other baselines can be compared, like Jiaxuan You, Rex Ying, and Jure Leskovec. Position-aware graph neural networks. In International Conference on Machine Learning, pp. 7134–7143. PMLR, 2019.
> >
> > **Authors**: We have reviewed the Position-aware GNN model in the related works, along with the limitations arising from the coupling of node representations with anchor-sets. We did not directly compare our approach to Position-aware GNN as their model was designed for link prediction tasks while our experiments focused on graph prediction tasks.
> >
> > Besides, our proposed architecture is quite different. Position-aware GNN relies on a set of anchor-sets to define positional features. There k-anchor sets of nodes are randomly selected for each graph and at each forward pass (meaning that for the same graph and two distinct forward passes, the set of anchors is different). Besides, the positional representation cannot be passed from a layer to the next one because the set of anchors is also different at each layer (once more randomly generated), and as such the positional embedding must be recomputed at each layer. Our proposed GNN does not rely on chosen anchors and the positional representation can be updated at each layer with the representation of the previous layer.
> > ___
> > >**Reviewer**: In summary, I think the proposed method is this work is very practical. It seems that using a simple positional embedding help a lot. However, it seems that the motivation is not very strong, like how to explain this positional embedding.
> >
> > **Authors**: We hope our answers have addressed your concerns. Let us know if you have any other questions or concerns.
> > ___
> > *Reference*:
> > [1] Chen et al., Can Graph Neural Networks Count Substructures? 2020

---

> ### Author Response · Authors · 2021-12-01
> **Follow-up for feedback on rebuttal**
>
> Dear Reviewer E8yv,
>
> We thank you again for taking your time reviewing this work. We put our best efforts to prepare the rebuttal to your questions. We would very much appreciate if you could engage with us with your feedback on our rebuttal. We would be glad to answer any further questions and clarify any concerns.
>
> Also, if you are satisfied with our answers, please consider revising your score.
>
> With best regards

---

### Official Review · Reviewer_uyjX · 2021-11-06

**Correctness:** 3
**Technical Novelty And Significance:** 2
**Empirical Novelty And Significance:** 2
**Recommendation:** 5
**Confidence:** 4

**Main Review:**

The paper discusses the important topic of how to better utilize the structure information in message-passing-based graph neural networks. The proposed structure and way to generate positional embeddings demonstrated good results on several datasets, however, it still lacks a thorough justification of where the improvement comes from and how each component contributes to the final quality.

## Pros
The paper tackles the problem of injecting positional embeddings in GNNs, which is an important topic of effectively utilizing structure information. The separate feed-forward network of RWPE demonstrates improved accuracies and can be a simple add-on to multiple GNN structures.

## Cons
1. The paper proposes two major components: random walk position embeddings and a standalone feed-forward network to update the embeddings. While I think these two strategies are a good add-on to many basic GNN structures, I do feel there are some points that the first part is of limited novelty, and some points are not properly justified in the arguments and experiments.
The added LapEig makes the two options of LapEig and RWPE vague in their motivations. In fact, the trace loss: Tr(p^T\Delta p) is enforcing the positional embeddings to resemble LapEig. While the authors claim that RWPE is better, I was wondering why having the positional embeddings in the “final” layer would contribute to a better output. If that would be the case, would simply appending these embeddings to the final layer give similar quality?
2. The RWPE itself is not novel, as the authors have also pointed that the idea is inherited from a previous. However, the feed-forward network's effectiveness is not properly proved. Specifically, it is not clear whether the gain attributes to the specific structure instead of the added parameters of the feedforward network. That is, suppose, we have the newly added position embeddings, but without the p_i^{l+1} updating mechanism, will we still have the boost of quality? Moreover, the added positional embeddings are also adding more capacity to the network.
Minor:
Typo: Proposed Architecture: first sentence: make easy -> make it easy
Why the edge dimension in Equation (2) is of dimension d?
The argument of flipping signs in LapPE is not the full story of Laplacian decomposition. In fact, the eigen-vectors of the same eigen-values can be any bases in that subspace (instead of different signs/directions).


**Summary Of The Paper:**

This paper proposes a framework that utilizes Random Walk Positional Embeddings (RWPE) as extra features to boost the performance of GNNs. In particular, positional embeddings are updated as a separate forward network in each layer. The framework has demonstrated improved quality by injecting its positional embeddings and feed-forward structures in several GNNs.


**Summary Of The Review:**

The paper discusses new position embeddings and the feed-forward networks. However, the novelty of the first part is limited and the effectiveness of the second part is not properly justified.

---

> ### Author Response · Authors · 2021-11-21
> **Response to Reviewer uyjX (Part 1/3)**
>
> We thank the reviewer for her/his careful reading and comments regarding our work. We would like to take this opportunity to justify our proposed method and clarify the key architecture components.
>
> >**Reviewer**: The proposed structure and way to generate positional embeddings demonstrated good results on several datasets, however, it still lacks a thorough justification of where the improvement comes from and how each component contributes to the final quality.
>
> **Authors**: The proposed framework can be justified both theoretically and experimentally.
>
> Theoretically, standard MP-GNNs have been proved to have a low representation power, and fail to distinguish simple graph structures. An approach to improve the GNN representation is to design k-order GNNs, which require the use of k-tuple of nodes and hence increase the complexity to O(n^k) while not necessarily improving the generalization performance. An alternative is to add positional encoding (PE) to the graph vertices, and it was shown in [Murphy et al., 2019] that the PE theoretically increase the representation power of any MP-GNN while keeping a linear complexity. This result serves as a theoretical foundation and justification of using PE in GNNs. But any PE would not work well s.a. the index PE of [Murphy et al., 2019], which has led us to propose two novel contributions in this line of work; (1) the use the diagonal of the k-step Random Walk Matrix as a discriminative and low-complexity PE (which leads to GNNs strictly more powerful than the 1-WL test as shown in Sections A.1 and A.2), (2) the update of PE at each layer to better adapt to the task at hand. This results in a very simple and powerful framework that can learn both structural and positional information and can be applied to any GNN.
>
> Experimentally, we refer to Section 4.3 (Table 3) where we inform the improvement of each component of the proposed LSPE framework. First, when we solely inject the PE (either LapPE or RWPE) into the GNN (GatedGCN) model, the performance improves from a MAE score of 0.251 to 0.202 and 0.122 for LapPE or RWPE respectively for ZINC. It shows numerically the importance of using positional encoding to improve the representation capability of GNN by disambiguating similar node structures in a graph and leading to a performance increase. This forms the premise of our work and previous studies have mostly used the approach to inject (different types of) PE in the first layer of a GNN model, see Eqns. (3-6) for an abstraction of MPGNNs-PE that encompasses such methods in the literature.
>
> In the above result, the gain provided by RWPE is much significant than the previously published LapPE. LapPE provides a unique representation of nodes. However, it is mathematically not well-defined as the sign of the eigenvectors are ambiguous, which forces the network to learn the sign invariance during training. Unlike LapPE, RWPE does not suffer from this sign ambiguity. Besides, this PE offers a unique node representation for nodes with distinct k-hop neighborhoods and RWPE is strictly more powerful than 1-WL (Section A.2). For these various reasons, the performance boost of RWPE is superior to LapPE. We refer to Sections A.1, A.2 and A.3 in the supplementary material for detailed discussions.
>
> Next, we observed that simply injecting RWPE into the first layer can lead to overfitting as the PE may not fit well to the task at hand in terms of test predictions. As such, the rationale of the proposed LSPE architecture is to update the positional representation along with the structural representation at every layer, Eqns. (7-9). This alleviates overfitting and improves the test MAE from 0.122 to 0.100. Finally, in Eqn. (12), the final layer fused the structural and positional representations together before the downstream loss, which allows to achieve the best MAE score of 0.093.
>
> To summarize the key components that lead to the proposal of our LSPE framework; (1) injecting of a PE (LapPE or RWPE) into the input layer (MAE score is 0.122 compared to No PE with 0.251), (2) updating PE at each layer along with the structural representation (MAE reduces to 0.100), and (3) fusing the structural and positional information (MAE final is 0.093). The above steps and their significance are presented in details in Section 4.3.
> __________

---

> > ### Author Response · Authors · 2021-11-21
> > **Response to Reviewer uyjX (Part 2/3)**
> >
> >
> > >**Reviewer**: The added LapEig makes the two options of LapEig and RWPE vague in their motivations. In fact, the trace loss: Tr(p^T\Delta p) is enforcing the positional embeddings to resemble LapEig.
> >
> > **Authors**: The most appropriate choice of PE naturally depends on the task and the dataset. For example, if we consider social graphs, then communities exist and LapPE, that use the k smallest eigenvectors, are known to be good indicators of the communities, and thus can provide meaningful PE. In the case of molecular graphs, where the community assumption is less valid, the use of RWPE seems more suitable as it provides a better indicator of the general graph topology.
> >
> > We agree that the Dirichlet loss, $Tr(p^T\Delta p)$, coupled with the orthogonality term, $||p^Tp-I||$, encourages the PE to resemble to LapPE but it is not strictly enforced as the positional energy represents a relaxation of the original eigendecomposition problem. The motivation to use this additional positional loss comes from the domain of Manifold Learning, where the goal is to embed the graph into a lower-dimensional space which can be meaningful and helps improving performance. Unfortunately, our experiments only show a slight improvement in the case of ZINC (from 0.093 to 0.090) and no improvement for the other datasets when using this positional loss. However, we thought that suggesting the use of a positional loss might of interest to the readers who could later define a better loss by leveraging ideas from Manifold Learning.
> > _____
> > >**Reviewer**: While the authors claim that RWPE is better, I was wondering why having the positional embeddings in the “final” layer would contribute to a better output. If that would be the case, would simply appending these embeddings to the final layer give similar quality?
> >
> > **Authors**: We ran the suggested experiment by the reviewer, that is simply appending the RWPE to the final layer with the structural representation. The MAE performance degrades significantly to 0.249 (vs. 0.093 with the proposed architecture). As such, it becomes critical to inject the positional information into the first layer, to inform as soon as possible the GNN of global geometric information of the nodes in the graph. This is analogous to Transformers for NLP, where the word positions in the sentence is supplied to the network in the first layer with the cosine and sine functions. Besides, updating the positional representation of nodes at each layer reduces overfitting better and increases the performance than simply injecting the PE at the input layer. This is why we also have the positional embedding at each layer, including the final layer.
> > ___
> >
> > >**Reviewer**: The RWPE itself is not novel, as the authors have also pointed that the idea is inherited from a previous.
> >
> > **Authors**: We agree that random walk-based methods or more generally graph diffusion techniques are not novel. Accordingly, we have listed the references and pointed to a previous work that we built upon. However, an important and practical difference to highlight for RWPE is that it has a low-complexity usage compared to the previous method [Li et al., 2020] that uses all pairwise random walk landing probabilities. This is critical when running the random walk for a large number of k steps to differentiate the nodes with different k-hop neighborhoods. For example, for molecular graphs, it is necessary to use a k value larger than 24 to uniquely represent most nodes of the molecules. It may not be possible to run [Li et al., 2020] with such a large k value because it would require to keep in memory 24 RW matrices, which would not be efficient or practical ([Li et al., 2020] actually uses a value k<=3).
> > ___
> > >**Reviewer**: However, the feed-forward network's effectiveness is not properly proved.
> >
> > **Authors**: We refer to our above answer about the usefulness of the disentangled feed-forward network to update positional representations, also discussed in Section 4.3. In addition, since the notion of position of nodes in a graph is ill-posed, the idea of initializing a PE based on a function of the original graph structure, and then tuning of the PE w.r.t. to the task at hand is presented in the paper, which turns out to be quite effective as demonstrated.
> > ___

---

> > > ### Author Response · Authors · 2021-11-21
> > > **Response to Reviewer uyjX (Part 3/3)**
> > >
> > > >**Reviewer**: Specifically, it is not clear whether the gain attributes to the specific structure instead of the added parameters of the feedforward network. That is, suppose, we have the newly added position embeddings, but without the p_i^{l+1} updating mechanism, will we still have the boost of quality? Moreover, the added positional embeddings are also adding more capacity to the network.
> > >
> > > **Authors**: To demonstrate that the gain is not attributed to adding more parameters to the network, we followed the same benchmarking protocol introduced in [Dwivedi et al., 2020] to fairly compare all models on ZINC. For ZINC, all the scores in Table 2a and Table 1 are produced with a fixed number of 500k model parameters. This allows to impartially compare all models, particularly the two models without and with LSPE; such as GatedGCN-NoPE (with a MAE of 0.251) and GatedGCN-RWPE-LSPE (with 0.093 MAE), and shows that the gain is not coming from additional parameters but rather from the new ideas of decoupling structural and positional information, using a powerful PE (with RWPE) and updating simultaneously positional and structural representation in the GNNs.
> > >
> > > To make sure the two models, GatedGCN-NoPE and GatedGCN-RWPE-LSPE, have the same number of parameters, we decreased the hidden feature dimension when LSPE is used (for example on ZINC, please refer to the difference in the ‘hidden_dim’ of GatedGCN-NoPE and GatedGCN-RWPE-LSPE in the ‘configs/’ directory of our source code commented anonymously in this forum. These are respectively 154 and 118 equalling model parameters in ~500k for both models enabling fair comparison).
> > >
> > > We mentioned the budget of 500k parameters for ZINC in the caption of Table 2, as well as the fact that the scores on OGBG-MOL in Tables 2b and 2c are taken from the OGB project and its leaderboard (Hu et al., 2020), where models have different number of parameters.
> > > ___
> > > >**Reviewer**: Minor: Typo: Proposed Architecture: first sentence: make easy -> make it easy
> > >
> > > **Authors**: Thank you. We updated this in the revised version of our manuscript.
> > > ___
> > > >**Reviewer**: Why the edge dimension in Equation (2) is of dimension d?
> > >
> > > **Authors**: For simplicity we consider ‘d’ as the hidden dimensions for each of h, p, e update. However, the hidden dimensions can be set arbitrarily without any issues.
> > > ___
> > > >**Reviewer**: The argument of flipping signs in LapPE is not the full story of Laplacian decomposition. In fact, the eigen-vectors of the same eigen-values can be any bases in that subspace (instead of different signs/directions).
> > >
> > > **Authors**: Absolutely, if some eigenvalues are identical then the associated eigenvectors are defined up to the orthogonal group. We thank the reviewer for this comment and we have updated the manuscript accordingly.
> > > ___
> > > **Authors**: We hope our answers have addressed your concerns. Let us know if you have any other questions or concerns.
> > > ___
> > > *References*:
> > > [1] Li, P., Wang, Y., Wang, H. and Leskovec, J., 2020. Distance Encoding: Design Provably More Powerful GNNs for Structural Representation Learning.
> > > [2] Murphy, R., Srinivasan, B., Rao, V. and Ribeiro, B., 2019, May. Relational pooling for graph representations.
> > > [3] Dwivedi, V.P., Joshi, C.K., Laurent, T., Bengio, Y. and Bresson, X., 2020. Benchmarking graph neural networks.
> > > [4] Hu, W., Fey, M., Zitnik, M., Dong, Y., Ren, H., Liu, B., Catasta, M. and Leskovec, J., 2020. Open graph benchmark: Datasets for machine learning on graphs.

---

> ### Author Response · Authors · 2021-12-01
> **Follow-up for feedback on rebuttal**
>
> Dear Reviewer uyjX,
>
> We thank you again for taking your time reviewing this work. We put our best efforts to prepare the rebuttal to your questions. We would very much appreciate if you could engage with us with your feedback on our rebuttal. We would be glad to answer any further questions and clarify any concerns.
>
> Also, if you are satisfied with our answers, please consider revising your score.
>
> With best regards

---

### Author Response · Authors · 2021-11-22
**Summary of the revised manuscript**

Dear Reviewers,

We would like to thank you very much for your time reading and evaluating this work.

We are grateful for the generally positive feedback about this work and its message we hoped to convey; that is a simple, generic and effective framework to improve the representation power and performance of any message-passing GNN by decoupling structural and positional information.

We have gladly answered your questions individually and point-by-point.

We have also revised and uploaded a new version of the manuscript based on the reviewers questions.
The summary of the main changes are as follows:
* Addition of 3 non-molecular graph datasets (IMDB-BINARY, IMDB-MULTI and CIFAR10) with their results and a discussion in supplementary Section D.
* Inclusion of an additional experiment in Table 3 (that compares different components leading to performance gain) by concatenating the structural and the positional RWPE only at the final layer.
* Adding the limitation of eigenvalue multiplicities in Section 2 (along with the discussion of Laplacian eigenvectors as PE).
* Revising the title “LSPE improves the state-of-the-art for domain-agnostic GNNs” in Section 4.2 to make it clear that the proposed technique did not include any domain expertise (although it is possible to do it, but we have not explored this direction).
* Correcting minor typos as suggested.

We hope that our answers and the revision of the manuscript clarified most of the concerns. If not, we will be happy to answer any additional questions.

With best regards

---

### Author Response · Authors · 2021-11-29
**Follow-up**

Dear Reviewers,

The discussion period is closing.

We would appreciate to know if you have any additional questions, concerns or clarifications you would like to ask us.

If we have addressed your concerns, please, consider revising your score.

We have put a lot of effort into this work and we also tried to write the best possible rebuttal to answer your questions.

Thank you very much for your understanding.

With best regards

---

### Decision · Program_Chairs · 2022-01-20

**Decision:**

Accept (Poster)

**Comment:**

This work adds the positional encoding (akin to those in transformers, but adapted) to GNNs.
In their reviews, reviewers raised a number of concerns about this work, in particular, lack of novelty, lack of ablations to demonstrate the claims of the paper, lack of comparison to previous work (e.g., position-aware GNNS, Graphormer and GraphiT which would appear very related to this work), lack of motivation (e.g., the introduced positional loss do not actually improve performance), and whether the experimental results were really significant.
During the rebuttal, the authors replied to the reviews, to address. the concerns that they could. Of the reviewers, unfortunately only one reviewer elected to respond to the authors. It is disappointing that the four other reviewers did not respond and overall the reviewers did not discuss this paper further.

The authors chose to highlight privately to the AC that two reviewers who scored the paper unfavourably did not respond. The authors then argued this should be taken into account in the score (presumably to make acceptance more likely)--however, two favourable reviewers also did not respond (not highlighted by the authors). I understand this kind of private request to the AC to dismiss unfavourable reviews (especially if they do not respond) is becoming common--I find it unhelpful--I can see who and who has not responded.

Nonetheless, looking at the responses to the original concerns of the reviewers highlighted above, I believe the authors have adequately addressed the concerns of the reviewers. Therefore i recommend acceptance but only as a poster.